# Dissociation between the critical role of ClpB of *Francisella tularensis* for the heat shock response and the DnaK interaction and its important role for efficient type VI secretion and bacterial virulence

Athar Alam[1], Igor Golovliov[1], Eram Javed[1], Rajender Kumar[1], Jörgen Ådén[2], Anders Sjöstedt[1]*

1 Department of Clinical Microbiology and Laboratory for Molecular Infection Medicine Sweden (MIMS), Umeå University, Umeå, Sweden, 2 Department of Chemistry, Umeå University, Umeå, Sweden

* anders.sjostedt@umu.se

**Data Availability Statement:** All relevant data are within the manuscript and its Supporting Information files.

## Abstract

*Francisella tularensis*, a highly infectious, intracellular bacterium possesses an atypical type VI secretion system (T6SS), which is essential for its virulence. The chaperone ClpB, a member of the Hsp100/Clp family, is involved in *Francisella* T6SS disassembly and type VI secretion (T6S) is impaired in its absence. We asked if the role of ClpB for T6S was related to its prototypical role for the disaggregation activity. The latter is dependent on its interaction with the DnaK/Hsp70 chaperone system. Key residues of the ClpB-DnaK interaction were identified by molecular dynamic simulation and verified by targeted mutagenesis. Using such targeted mutants, it was found that the *F. novicida* ClpB-DnaK interaction was dispensable for T6S, intracellular replication, and virulence in a mouse model, although essential for handling of heat shock. Moreover, by mutagenesis of key amino acids of the Walker A, Walker B, and Arginine finger motifs of each of the two Nucleotide-Binding Domains, their critical roles for heat shock, T6S, intracellular replication, and virulence were identified. In contrast, the N-terminus was dispensable for heat shock, but required for T6S, intracellular replication, and virulence. Complementation of the Δ*clpB* mutant with a chimeric *F. novicida* ClpB expressing the N-terminal of *Escherichia coli*, led to reconstitution of the wild-type phenotype. Collectively, the data demonstrate that the ClpB-DnaK interaction does not contribute to T6S, whereas the N-terminal and NBD domains displayed critical roles for T6S and virulence.

## Author summary

Type VI secretion systems (T6SSs) are essential virulence determinants of many Gram-negative pathogens, including *Francisella tularensis*. This highly virulent bacterium encodes an atypical T6SS lacking ClpV, the ATPase crucial for prototypic T6SS sheath

**Funding:** We acknowledge research funding for this work by grants 2013-4581 (to AS) and 2013-8621 (to AS) from the Swedish Research Council and a Biotechnology grant (FS 2.1.6-2291-18 to AS) from the Medical Faculty, Umeå University, Umeå, Sweden, and the JC Kempe Memorial Foundation (JCK-1624 to AS). The funders had no role in study design, data collection and analysis, decision to publish, or preparation of the manuscript.

**Competing interests:** The authors have declared that no competing interests exist.

disassembly. It, however, possesses ClpB, a protein critical for heat shock survival via its interaction with DnaK. Since ClpB possesses ATPase activity, it has been hypothesized to provide a compensatory function for the absence of ClpV, a hypothesis supported by the recent findings from us and others. Here, we investigated how *F. tularensis* ClpB controls T6S. *In silico* modelling of the ClpB-DnaK complex identified key interactions that were experimentally verified. For example, mutating one of the DnaK-interacting residues rendered the bacterium exquisitely susceptible to heat shock, but had no effect on T6S and virulence. In contrast, removing the N-terminal of ClpB only had a slight effect on the heat shock response, but strongly compromised both T6S and virulence. Intriguingly, the *Escherichia coli* ClpB could fully complement the function of *F. tularensis* ClpB. The data demonstrate that the two critical roles of ClpB, mediating heat shock survival and effective T6S, are dissociated and that the N-terminal is crucial for T6S and virulence.

## Introduction

The zoonotic disease tularemia is caused by the extremely virulent, facultative intracellular Gram-negative coccobacillus *Francisella tularensis* [1] and the subsp. *tularensis* and *holarctica* are important human pathogens. The related species *F. novicida* is a very rare human pathogen, but still highly virulent for mice, and therefore, commonly used as a laboratory model for tularemia [2]. The pathogenicity of both *Francisella* species is linked to the *Francisella* Pathogenicity Island (FPI), a gene cluster encoding a functional, but atypical type VI secretion system (T6SS) [3, 4].

Recently, it has been demonstrated that the FPI of *Francisella* encodes a functional T6S system, despite that individual components demonstrate low sequence similarity to canonical T6SS proteins, and it also lacks the two ATPases, IcmF/TssM and ClpV, both of which are believed to provide the energy for secretion in prototypical T6SS [5–9]. The ClpV homologue is completely absent in the *Francisella* genomes; however, an IcmF homologue, termed PdpB, is present, but missing the Walker A motif necessary for the ATPase activity [10]. The ClpV homologue is also absent in some other species, such as *Campylobacter jejuni*, *Helicobacter hepaticus*, and *Salmonella choleraesuis*, but these species still demonstrate functional T6S, indicating that ClpV is not vital for all species [11, 12]. Furthermore, only a partial loss of the function of T6SS was observed in a *V. cholerae* Δ*clpV* strain, demonstrating that ClpV is an important, yet nonessential component of T6SS in *V. cholerae* [13, 14]. In the absence of ClpV, it is possible that other proteins, such as ClpB, due to its ATPase activity, may contribute to the assembly-disassembly cycle of the T6S apparatus.

Recently, it was reported that ClpB co-localizes with the contracted IglA/IglB sheaths of *F. novicida* T6SS [6] and, moreover, we reported that ClpB mutants of *F. tularensis* subspecies *holarctica* and *tularensis* are defective for type VI secretion [5]. The ClpB protein is conserved between *F. novicida* and *F. tularensis*. ClpB has been shown to be required for virulence of many bacteria including *F. tularensis*, *Porphyromonas gingivalis*, *Mycoplasma pneumoniae*, *Listeria monocytogenes*, *Piscirickettsia salmonis*, and *Mycobacterium tuberculosis* [5, 15–19].

ClpB belongs to the ring-forming Clp/Hsp100 proteins family and confers heat shock survival to a range of species via its unfoldase activity, a role executed jointly with the co-chaperones DnaJ, DnaK, and GrpE [20, 21],[22–24]. Out of these, only DnaK physically interacts with ClpB. Upon being recruited to the substrate by DnaJ, DnaK hydrolyses ATP, which allows the formation of a stable substrate-DnaK complex. The complex is brought to the substrate-processing central pore of ClpB through direct interaction between the nucleotide-binding

domain (NBD) of DnaK and the coil-coiled domain of ClpB [25]. Once unfolded substrates are recovered, GrpE facilitates the exchange of the DnaK-bound nucleotide, thereby triggering the release of the substrate and allowing DnaK to participate in another round of substrate binding [25, 26].

ClpB, a member of class 1 AAA+ proteins, consists of two AAA (ATPases Associated with diverse cellular Activities)-domains, NBD 1 and 2, each containing the Walker-A, Walker-B, and arginine finger motifs, flanked by an N-terminal and a C-terminal domain and separated by a central linker domain [23]. Inserted in NBD-1 is a long coiled-coil middle domain (M-domain) that distinguishes ClpB from other Hsp100 proteins. The M-domain is critical for interactions with the Hsp70 chaperone system and facilitates protein disaggregation [27], whereas the N-terminal is proposed to have a regulatory role in substrate recognition and protein disaggregation [20, 28]. The NBDs have been shown to bind and hydrolyse ATP that stabilizes the ClpB hexamers and its interaction with the substrate [29]. Although much detailed information is available for ClpB from several bacterial species, an understanding of the roles of the conserved domains of *Francisella* ClpB is lacking.

In the present study, we demonstrate that the Δ*clpB* mutant of *F. novicida* is exquisitely susceptible to heat shock, shows defective intracellular growth and markedly decreased virulence in the mouse model, concomitantly with impaired T6S. We observed that substituting an amino acid critical for the DnaK interaction led to extreme heat shock susceptibility of the resulting mutant; however, it showed intact T6S and virulence. In contrast, an N-terminal mutant demonstrated a normal heat shock response, but defective T6S and virulence. Our findings reveal that the crucial roles of the *F. novicida* ClpB for heat shock and regulation of T6S are dependent on distinct regions of the protein and not dependent on the DnaK interaction.

## Results

### The M-domain of ClpB interacts with the subdomain IB and IIB of DnaK

A vital role of ClpB in T6S of *Francisella* has been established [5, 6]; however, whether this depends on the ability of ClpB to interact with DnaK of the Hsp70 chaperone system is unknown. We therefore employed a computational approach to identify ClpB residues critical for the DnaK interaction. The models for ClpB and DnaK were created based on the crystal structure of *Thermus thermophilus* ClpB [30] (Fig 1A) and the *E. coli* DnaK (DnaK$_{EC}$), respectively (Fig 1B) [31]. *F. novicida* ClpB-DnaK was docked and the best-fit conformation, based on their key interactions and the scores, was selected for further analysis. The analysis indicated that subdomains IB and IIB of DnaK interact with the M-domain of ClpB (Fig 1C), which also has been observed in the ClpB$_{EC}$-DnaK$_{EC}$ model [26]. Since this initial ClpB-DnaK complex obtained by information-driven flexible docking approach was not fully flexible, the explicit-solvent molecular dynamics (MD) simulations were performed up to 100 ns with further refinement of the best docking conformation. Significant conformational changes of both proteins were observed during the entire simulation (S1 Video) and a total of 46 different combinations of interactions between the ClpB and the DnaK residues with occupancy scores ranging from 0.05% to 42.8% were observed (S3 Table). When the low interaction combinations were excluded by keeping the hydrogen-bonding interaction occupancy cutoff at 5%, a total of 10 residues from DnaK and 6 residues from ClpB was found to be involved in a total of 11 different interaction combinations. These included subdomain IB residues R56 and Q57 and subdomain IIB residues K258, Q262, R263, E266, E269, N284, Y287, and H297 of DnaK, and the M-domain motif-2 residues S496, E500, Q502, Y503, E508, and E510 of ClpB (Fig 1D). The M-domain motif-2 of ClpB$_{EC}$ has been reported to act as a platform for the DnaK$_{EC}$-

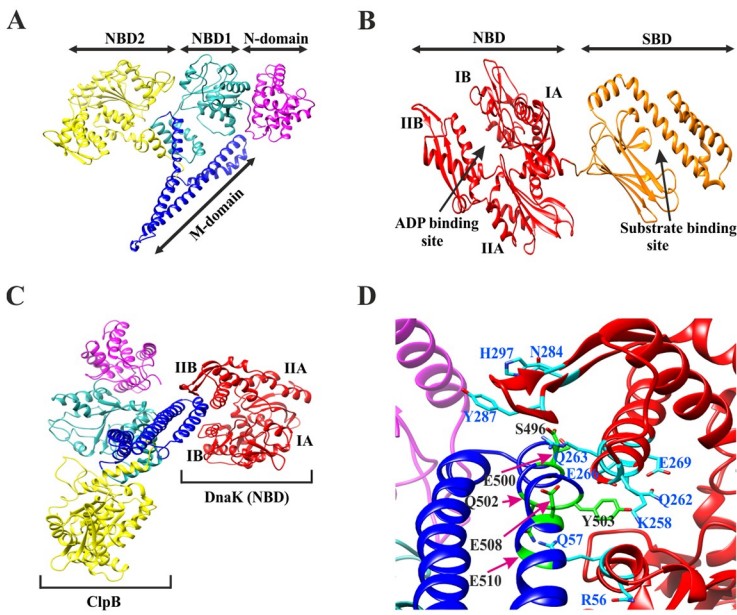

**Fig 1. Model structures of ClpB, DnaK and the molecular dynamic simulation.** (A) The overall ClpB monomer structure, comprised of an N-domain (magenta), nucleotide binding domain-1 (NBD-1) (light blue), NBD-2 (yellow) and M-domain shown (blue). The image was generated using the UCSF Chimera program based on the *T. thermophilus* ClpB. (B) The model structure of DnaK is comprised of an N-terminal NBD (red) containing four subdomains, IA, IB, IIA and IIB, and a substrate-binding domain (SBD) (orange). The image was generated using the UCSF Chimera program based on the *E. coli* DnaK. (C) The best-docked complex of ClpB$_{Ft}$-DnaK$_{Ft}$. The DnaK subdomains IB and IIB are in contact with the ClpB M domain. (D) An average complex structure of ClpB$_{Ft}$-DnaK$_{Ft}$ during 100 ns molecular dynamic (MD) simulations. Important hydrogen bonding interactions between ClpB residues (black) and DnaK residues (light blue) observed during the entire MD simulation are highlighted.

ClpB$_{EC}$ interaction [32]. All residues, except M-domain motif-2 residue S496 of ClpB and subdomain IIB residues K258 and E266 of DnaK, are evolutionary highly conserved (S1A and S1B Fig). The model corresponds well to recent results that identified that subdomain IB residues R56 and Q57 of DnaK$_{EC}$ (corresponding to the same residues of *F. novicida* DnaK) and the subdomain IIB residues R261 and N282 of DnaK$_{EC}$ (corresponding to residues R263 and N284, respectively, of *F. novicida* DnaK) are involved in the reactivation of heat-denatured proteins; however, only the residues of subdomain IIB contributed directly to the DnaK$_{EC}$ and ClpB$_{EC}$ interaction [26]. Similarly, another study identified amino acids 456–520 of motif 2 of the M-domain of ClpB$_{EC}$ (corresponding to same residues of *F. novicida* ClpB) to be involved in the interaction with DnaK$_{EC}$ [32].

## Mutations in helix-2 of the ClpB M-domain affect interaction with the DnaK chaperone system, disaggregation activity, and ATPase activity *in vitro*

To validate the bioinformatic data, we generated substitution mutants of all aforementioned M-domain amino acids of ClpB, except the highly variable S496 (S1A Fig), and assessed the unfolding activities of the corresponding mutants. We first tested the impact of M-domain variants of ClpB on the disaggregation of aggregated malate dehydrogenase (MDH), or firefly luciferase as model substrates *in vitro*. Both assays require the physical interaction of ClpB and DnaK. Solubilization of MDH aggregates was monitored by determining the decrease in sample turbidity. In the presence of the chaperones DnaK, DnaJ, and GrpE (KJE), ClpB variants E500A and E508A displayed 40–45% of wild-type levels of disaggregation, whereas Q502A resulted in very modest change of disaggregation, not different from the background

level (Fig 2A). Y503A, which showed the highest occupancy score with DnaK and engaged as a sandwich between the domain IB and IIB of DnaK in the MD simulation (Fig 1D, S1 Video), also resulted in very modest disaggregation, not different from the background level (Fig 2A). The data with Y503A is in agreement with previously published data on other bacterial species demonstrating that the mutation of Y503A abolished the DnaK interaction [25, 32, 33]. E510A, on the other hand, resulted in the same change of turbidity as did wild-type ClpB. A similar effect was observed when using aggregates of urea-denatured luciferase; however, the changes were of slightly higher magnitude than those observed with MDH. Differences in the size and structure of the MDH and luciferase aggregates could explain these differences. While Y503A activity was only slightly above that of the negative control (21%

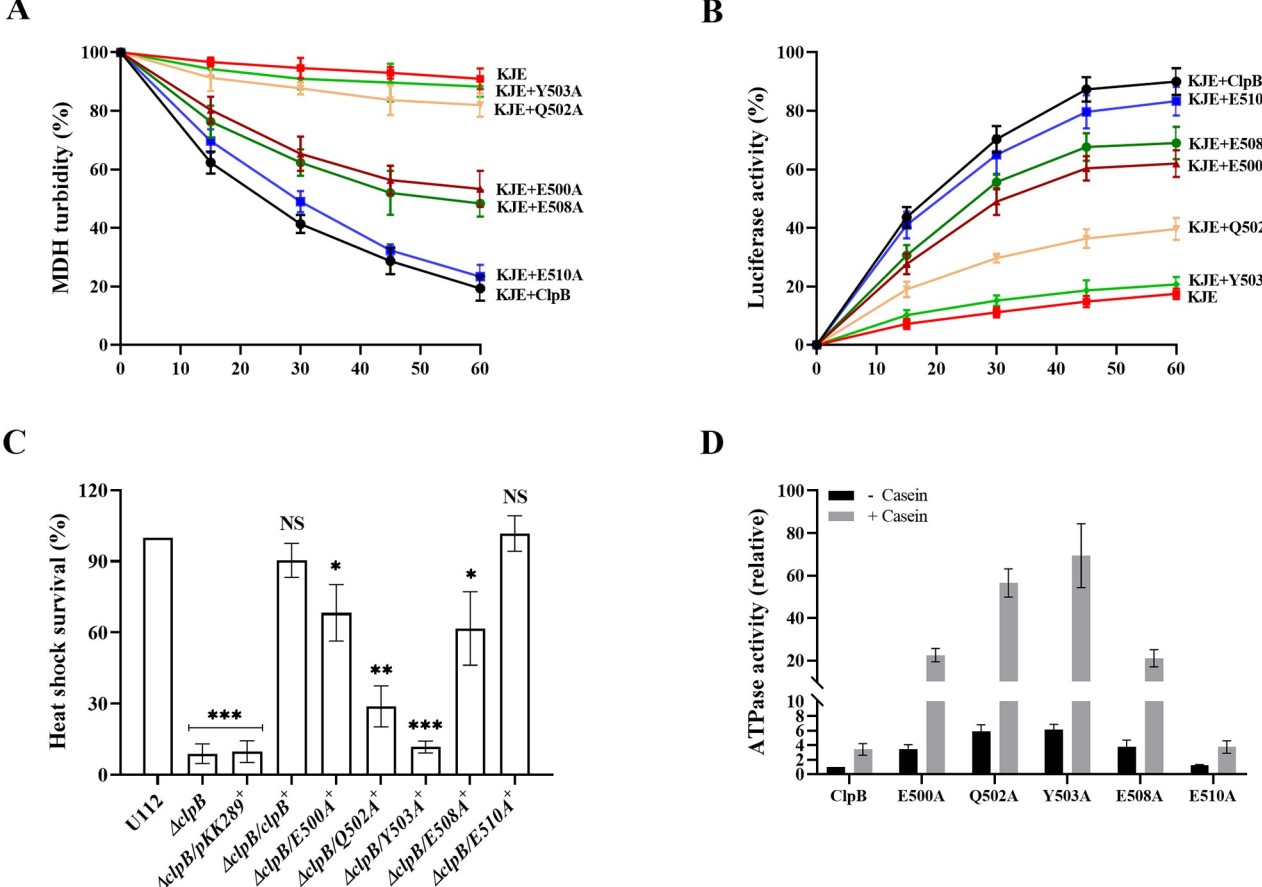

**Fig 2. Characterization of M-domain variants of ClpB with regard to disaggregation and ATPase activity.** (A) MDH disaggregation activities of M-domain variants of ClpB were monitored by loss of turbidity in the presence of ATP and the co-chaperones DnaK, DnaJ and GrpE (KJE) of *F. novicida*, as described in Materials and Methods. The initial MDH turbidity was set as 100% and data were calculated compared to the denatured MDH and shown as percentage of disaggregation. At least three independent experiments were performed and data with mean ± SD are shown. (B) ClpB-mediated refolding activities of urea-denatured luciferase were determined in the presence of ATP and the co-chaperones KJE of *F. novicida*, as described in Materials and Methods. Refolding results in an increase of fluorescence over time. The KJE control refers to co-chaperones only (no ClpB). The initial fluorescence was set as zero and data of at least three independent experiments with mean ± SD are shown. (C) Heat shock survival of indicated *F*: *novicida* strains upon heat shock. Bacteria were exposed to 50°C for 30 min and the mean ± SD CFU are indicated. The wild-type strain U112 did not exhibit any significant killing during the treatment, and the value was set as 100%. Sign + indicates *trans* complemented strains. A significant difference in the bacterial numbers of mutant strains *vs*. U112 is indicated as follows: *** $P < 0.001$; ** $P < 0.01$; * $P < 0.05$; NS (not significant) $P > 0.05$. (D) ATPase activity of wild-type and indicated M-domain variants of ClpB were determined in the absence or presence of α-casein (10 mM). Basal ATPase activity of wild-type ClpB was set at 1.0. At least three independent experiments were performed and data with mean ± SD are shown.

activity), disaggregation activities of E500A (61%), E508A (68%), and Q502A (38%) were distinctly higher and in the case of the former two, intermediate between that of the negative control and that of wild-type ClpB (83%) (Fig 2B). To determine whether the mutants had preserved hexameric conformation, we analyzed the purified proteins with gel filtration. None of the ClpB mutants showed defects in hexamer assembly as revealed by size exclusion chromatography (S2 Fig). Potential alterations in the chaperone activities of the ClpB variants can therefore be directly attributed to the introduced mutations.

To further extend our investigation, M-domain variants of *clpB* were expressed *in trans* in the Δ*clpB* background and the phenotypes of the resulting mutants were characterized with regard to their susceptibility to heat shock. After 30 min at 50˚C, the CFU of the wild-type U112 strain did not change significantly and this level was denoted as 100%. The Δ*clpB*/*E500A*$^+$ and Δ*clpB*/*E508A*$^+$ showed 58–66% survival, whereas the survival of Δ*clpB*/*Q502A*$^+$ was 32%. Δ*clpB*/*Y503A*$^+$, which showed almost no activity in disaggregation assays, demonstrated levels similar to the vector control, <10% of that of Δ*clpB*/*clpB*$^+$ (Fig 2C). As expected, Δ*clpB*/*E510A*$^+$ showed survival similar to that of Δ*clpB*/*clpB*$^+$, corroborating the results of the disaggregation assays. It is important to note that the *in trans* expression of each mutant was more than 20-fold higher than that of the wild-type strain U112 (S3 Fig), but specific mutants still conferred limited, or no heat shock survival compared to Δ*clpB*, indicating that the expression level of ClpB is not critical for the heat shock survival, but rather the degree of interaction with DnaK.

To further dissect the regulatory role of the M-domain variants of ClpB for the disaggregation and stress response, we investigated whether the mutations affected the ATPase activity of ClpB. The M-domain of ClpB is of special relevance in this regard, since it negatively regulates the substrate-stimulated ATPase activity of ClpB and ultimately disaggregation of aggregated proteins [25, 32–34]. The specific ATPase activities of wild-type ClpB and the M-domain mutants were determined in the absence or presence of the model substrate α-casein. The ATPase activity for wild-type ClpB was 3.4-fold higher in the presence of the substrate (Fig 2D). In comparison to the wild-type, the ATPase activities of the substitution mutants in the absence or the presence of substrate were as follows; E500A (3.4-fold, 6.4-fold), Q502A (5.9-fold, 16.4-fold), Y503A (6.1-fold, 20.1-fold), E508A (3.7-fold, 6.1-fold), and E510A (1.3-fold, 1.1-fold), and (Fig 2D). Thus, all the M-domain variants in comparison to the wild-type, except E510A, exhibited similar patterns with regard to ATPase activity; a much increased basal rate of ATP hydrolysis and also increased activity in the presence of α-casein.

Collectively, the data indicate that, similar to what has been shown for ClpB of other species, [25, 32–34], the M-domain of the *Francisella* ClpB negatively regulates ATPase activity.

## The ClpB-DnaK interaction is dispensable for intracellular replication, T6S and bacterial virulence

To further extend our analysis, we investigated the effects of *in trans* complemented M-domain variants of *clpB* on intracellular replication in the murine macrophage cell line J774A.1 and for KCl-induced substrate secretion, a method previously used to evaluate T6S of *F. novicida* [7, 35]. The secretion was assessed by measuring the amount of IglC secreted into the culture supernatant *in vitro*. Unlike the distinct differences observed in the disaggregation assays and the heat shock survival, all Δ*clpB* complemented with the M-domain variants of *clpB* showed similar intracellular replication (Fig 3A) and KCl-induced substrate secretion (Fig 3B) with no significant differences compared to the wild-type.

As aforementioned, *in trans* complemented *clpB* variants express ClpB many fold higher than the wild-type (S3 Fig) and this may have influenced the intracellular replication and substrate secretion measured. To investigate the effect of the protein expression in general, we

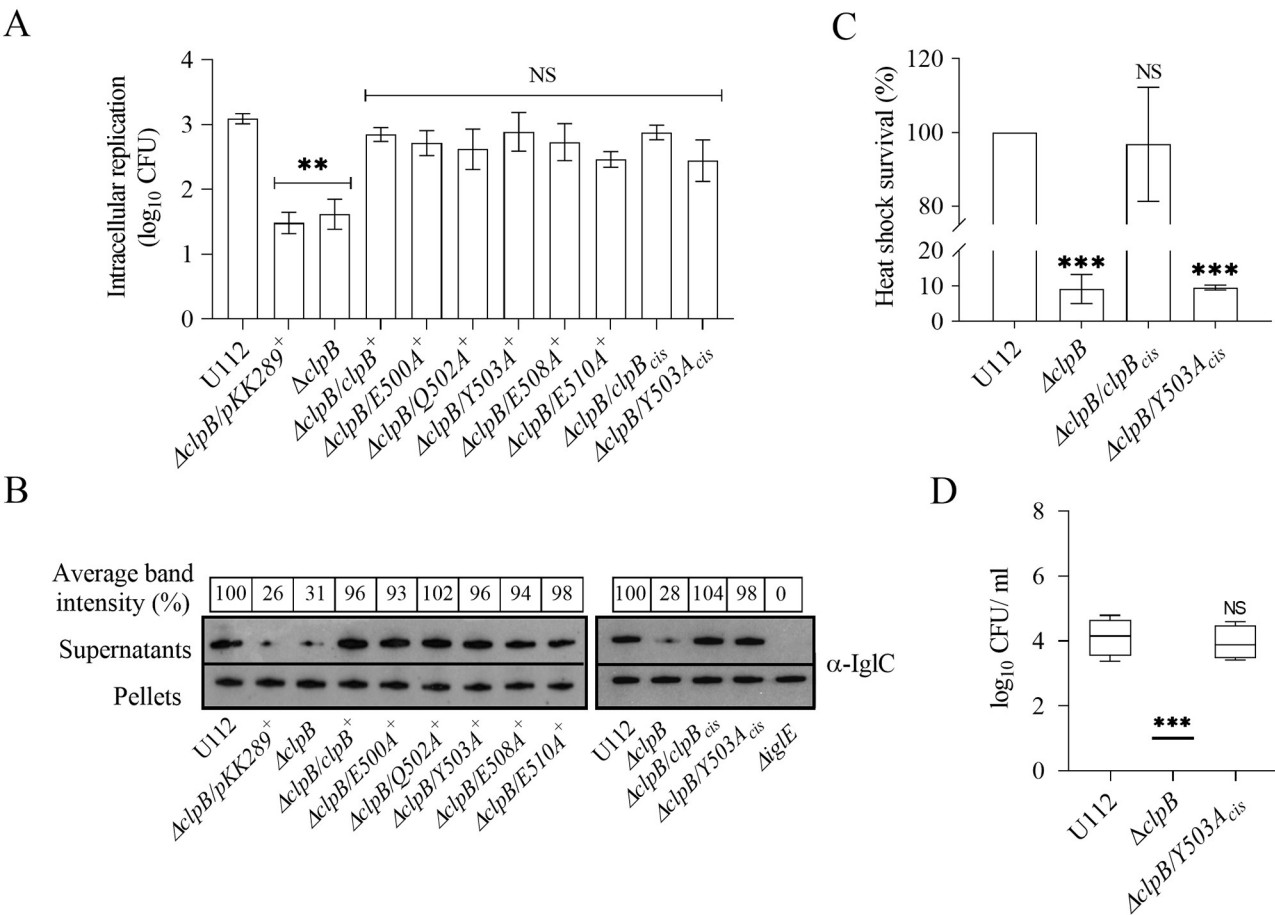

**Fig 3. Characterization of M-domain variants of ClpB with regard to intracellular replication, substrate secretion and heat shock survival.** (A) The wild-type strain U112, Δ*clpB*, or Δ*clpB* expressing M-domain variants of *clpB* variants in *trans* or inserted in *cis* on the chromosome, were used to infect J774A.1 cells. Infected cells were lysed at 0 h and 24 h and the number of CFU were determined. The net growth mean values ± SEM of at least three independent experiments are shown. A significant difference in the bacterial numbers of mutant strains *vs.* U112 is indicated as follows: ** $P <$ 0.01; NS (not significant) $P > 0.05$. (B) Analysis of T6S by bacterial strains. Indicated strains were grown at 37°C to an OD of 1.5 in TSB medium supplemented with 5% KCl. Precipitated supernatants or pellets of the same strain were separated by SDS-PAGE and analyzed using Western blot analysis and anti-IglC antiserum. The signal intensity of each band on scanned images of supernatant samples was measured using the Image J program (http://rsbweb.nih.gov/ij/) and the signal of each strain is presented as a percentage of the band-intensity of the U112 strain, the latter set as 100%. The in-frame deleted Δ*iglE* mutant of *F. novicida* U112 was used as negative control. At least three independent experiments were performed and a representative image with the average band intensity percentage is shown. (C) Survival of indicated strains upon heat shock. Bacteria were exposed to 50°C for 30 min and the (mean ± SD) CFU are indicated. The wild-type strain U112 did not exhibit any significant killing during the treatment, and the value was set as 100%. A significant difference in the bacterial numbers of mutant strains *vs.* U112 is indicated as follows: *** $P < 0.001$; ** $P < 0.01$; NS (not significant) $P > 0.05$ for *cis*-complemented strains *vs.* U112. (D) After subcutaneous inoculation with $1 \times 10^3$ CFU of the indicated *F. novicida* strains, mice were sacrificed on day 3, and bacterial burdens ($\log_{10}$ CFU/ml) in liver were determined. The mean ± SEM for six mice per group is indicated. A significant difference in the bacterial numbers of mutant strains *vs.* U112 is indicated as follows: *** $P < 0.001$; NS (not significant) $P > 0.05$.

complemented Y503A, the most defective mutant, *in cis*, designated as Δ*clpB/Y503A*$_\text{cis}$, and analyzed the heat shock survival, intracellular replication, and KCl-induced substrate secretion. Expression of the *Y503A*$_\text{cis}$ did not rescue the highly susceptible heat shock phenotype of the *clpB* mutant (Fig 3C), whereas the intracellular replication (Fig 3A) and KCl-induced IglC secretion (Fig 3B) were very similar to the wild-type strain, further strengthening the conclusion that the ClpB-DnaK interaction, but not the level of ClpB expression, is vital for heat shock survival, whereas the interaction is dispensable for intracellular replication and KCl-induced substrate secretion.

To determine if the ClpB-DnaK interaction is also vital for virulence, mice were subcutaneously infected and the spread of *F. novicida* U112, Δ*clpB*, and Δ*clpB*/*Y503A*$_{cis}$ was followed by determining the bacterial numbers on day 3 and 5 in spleen and liver, the main target organs of tularemia. Δ*clpB*/*Y503A*$_{cis}$ showed no significant attenuation regardless of organ and time point compared to the U112 strain, whereas Δ*clp*B demonstrated very significant attenuation and bacterial numbers were barely above the detection limit. Similar results were observed in three independent experiments, and the data for bacterial counts in liver on day 3 are shown (Fig 3D). The results demonstrate that expression of ClpB *per se* is critical for intracellular replication, substrate secretion, and virulence, whereas its interaction with DnaK is dispensable for the same phenotypic traits.

## Mutation of conserved residues within the NBDs of ClpB severely affects substrate secretion and bacterial virulence

ClpB comprises two NBDs (NBD-1 and NBD-2), which bind and hydrolyse ATP. The cooperative action of the ClpB and the DnaK system of *E. coli* requires both NBDs to be functional for disaggregation of aggregated proteins and heat shock survival [36]; however, their role in substrate secretion and bacterial virulence is unknown. We asked if the NBDs of *F. novicida* ClpB play any role in T6S and bacterial virulence by generating a series of single Walker A motif mutants of NBD-1 (K212A, designated WA1) and NBD-2 (K613A–WA2), Walker B motif mutants of NBD-1 (E279A–WB1) and NBD-2 (E680A–WB2), Arginine finger of NBD-1 (R332A–Arg1) and NBD-2 (R757A–Arg2), and mutants in both NBD-1 and NBD-2 (*i.e.*, WA1-2, WB1-2, and Arg1-2) based on the conservation analysis and multiple sequence alignment (S1A and S4 Figs) and tested their effects on the ClpB function.

We first analyzed their ability to replicate intracellularly within murine BMDM. All the Δ*clpB* complemented variants showed similar intracellular replication, not significantly different compared to the Δ*clpB* mutant, but significantly lower than U112 (Fig 4A). Growth could be restored to the wild-type level by expressing wild-type *clpB in cis* (Fig 4A). The heat shock survival of these strains were assessed and the viability decreased by 97% for the Δ*clpB* mutant when exposed to heat shock ($P < 0.001$; Fig 4B), while the CFU of the wild-type strain U112 did not change significantly. The survival of all three double substitution mutants (WA1-2, WB1-2, and Arg1-2) was severely affected and comparable to the Δ*clpB* mutant, whereas the single substitution mutants showed variable levels of survival, but strongly impaired compared to U112 ($P < 0.001$; Fig 4B).

When tested for the KCl-induced substrate secretion, the Δ*clpB* mutant showed much impaired secretion, ~28% of the level of the wild-type U112 strain, whereas IglC secretion was restored when *clpB* was expressed *in cis* (Fig 4C). In comparison, levels for a prototypical FPI mutant, the *vgrG* transposon mutant (vgrG::Tn), was below the detection limit (Fig 4C). All substitution mutants showed similar level of IglC secretion, ranging from 25–41%, thus, close to the level of Δ*clpB* and much impaired compared to ClpB (Fig 4C).

Since NBDs play vital roles in the oligomerization and ATPase activity of ClpB [37], we further characterized the oligomeric status of the ClpB mutants and their ATPase activity. Most ClpB variants formed hexamers (S2 Fig), with the exception of WA1, WA1-2, Arg1 and Arg1-2 that all formed dimers. When the ATPase activity of the ClpB mutants was measured, the double mutants, WA1-2, WB1-2 and Arg1-2, demonstrated no activity, not even in presence of the substrate α-casein (Fig 4E). The basal ATPase activity of WA1 was approximately 30% of wild-type levels, whereas the activities of WA2, WB1, and Arg-1 were similar to that of the wild-type (Fig 4E). In the presence of α-casein, the activities of WA1, WB1, and Arg-1 were similar to that of the wild-type protein, while that of WA2 was slightly higher (Fig 4E). The

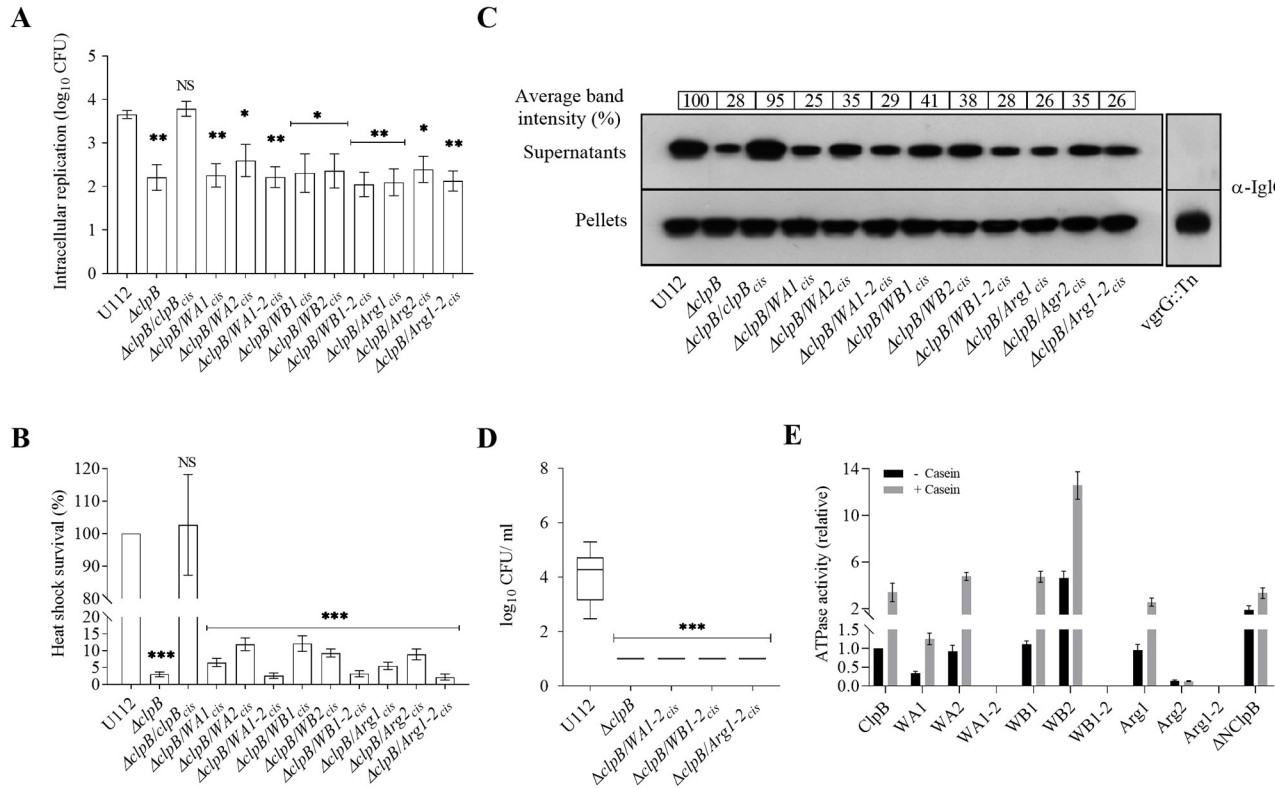

**Fig 4. Characterization of Walker and Arginine finger motif mutants of *F. novicida*.** (A) The wild-type strain U112, Δ*clpB* or Δ*clpB* expressing various ClpB variants of NBD-1 (WA1, WB1 or Arg1), NBD-2 (WA2, WB2 or Arg2) or both NBDs (WA1-2, WB1-2 or Arg1-2) *in cis* were used to infect BMDM. The specific mutations were as follows WA1: K212A, WB1: E279A, Arg1: R332A, WA2: K613A, WB2: E680A, and Arg2: R757A. Infected cells were lysed at 0 h and 24 h and the number of CFU were determined. The net growth mean values ± SEM of at least three independent experiments are shown. A significant difference in the bacterial numbers of mutant strains *vs.* U112 is indicated as follows: ** $P < 0.01$; * $P < 0.05$; NS (not significant) $P > 0.05$ (B) Survival of indicated strains upon heat shock. Bacteria were exposed to 50°C for 30 min and mean ± SD CFU are shown. The wild-type strain U112 did not exhibit any significant killing during the treatment, and this value was set as 100%. A significant difference in the bacterial numbers of mutant strains *vs.* U112 is indicated as follows: *** $P < 0.001$; NS (not significant) $P > 0.05$. (C) Analysis of T6S of bacterial strains. Indicated strains were grown at 37°C to an OD of 1.5 in TSB medium supplemented with 5% KCl. Supernatants were collected, filter sterilized and TCA precipitated. Precipitated supernatants or pellets of the same strain were separated by SDS-PAGE and analyzed using Western blot analysis with an anti-IglC antiserum. The signal intensity of each band was measured as described in Fig 3B and percentage of the band-intensity *vs* U112 (set as 100%) is presented. At least three independent experiments were performed and a representative image is shown. (D) After subcutaneous inoculation with $1 \times 10^3$ CFU of the indicated *F. novicida* strains, mice were sacrificed on day 3, and bacterial burdens ($\log_{10}$ CFU/ml) in liver were determined. The mean ± SEM for six mice per group is indicated. A significant difference in the bacterial numbers of mutant strains *vs.* U112 is indicated as follows: *** $P < 0.001$. (E) ATPase activity of wild-type ClpB, NBDs variants, and the N-terminal truncated (ΔNClpB) variant of ClpB was determined in the absence or presence of α-casein (10 mM). Basal ATPase activity of wild-type ClpB was set at 1.0. Experiments were conducted in triplicate and mean ± SD are shown.

activity of WB2 was significantly higher than that of the wild-type with or without α-casein, whereas the Arg2 mutant showed an almost complete loss of ATPase activity. Compared to the wild-type, ΔNClpB had a slightly increased basal ATPase activity in the absence of casein, but their activities were similar in the presence of α-casein (Fig 4E). The findings regarding the N-terminal mutant are in agreement with data obtained for the corresponding mutant of *E. coli* [37].

The observed loss of functions for several of the mutants could be due to conformational changes, protein instability, or a general defect in oligomerization, since each of these properties are potentially important for the ClpB activity. The analysis indicated that WA1, WA1-2, Arg1, and Arg1-2 could not form hexamers and, thus, the lack of normal conformation could explain their phenotypic defects. However, for the other mutants, our bioinformatic prediction

(S4 Table), CD analysis (S5A Fig), and the Western blot analysis of the protein levels (S5D Fig) indicated that they exhibited no major changes in conformation or stability, thus, the most logical explanation for their phenotypic defects is the specific amino acid mutation within each motif. Notably, several of the single NBD mutants showed an ATPase activity similar to or higher than ClpB, demonstrating that there are complementary functions of each domain. The notable exceptions were Arg2 and WB1-2 that both showed intact conformation and stability, but essentially abolished ATPase activity. This demonstrates that, despite a hexameric conformation, a double mutant will not possess any complementary function and therefore loses ATPase activity. The loss of activity of Arg2 has precedence since the corresponding mutant of *E. coli* is also defective for ATPase activity.

In conclusion, the introduction of mutations in each of the three motifs in both NBD domains resulted in much impaired heat shock survival, intracellular replication, and T6S, similar to the level observed for Δ*clpB*, despite variable effects on their ATPase activity.

## Mutations in the NBDs of *clpB* markedly affect the virulence of *F. novicida*

The Δ*clpB* mutants of SCHU S4, LVS, and *F. novicida* are highly attenuated and the behavior of the SCHU S4 Δ*clp*B mutant has been studied in much detail [5, 38]. The attenuated phenotypes observed for the mutants of conserved residues in NBDs with respect to heat shock survival, intracellular replication and T6S suggested that they would also show attenuated phenotypes *in vivo*. Mice were subcutaneously infected with *F. novicida* U112, Δ*clpB*, or the three double mutants, Δ*clpB*/*WA1-2cis*, Δ*clpB*/*WB1-2cis*, and Δ*clpB*/*Arg1-2cis*, and bacterial numbers determined in spleen and liver on day 3 and 5. All mutants showed much lower bacterial numbers in both organs and at both time points than the U112 strain, the differences ranged from 2–5 $\log_{10}$ and the numbers for the mutants were barely above the detection limit. Similar results were observed in three different experiments and the data for the CFU in the liver on day 3 are presented (Fig 4D). To confirm that the effect of the *clpB* mutation was specifically related to T6S and not to a general defect in FPI protein expression, Western blot analysis using anti-FPI antibodies against different FPI proteins was performed. No differences were observed between the lysate of the wild-type U112 and the respective Δ*clpB* mutant, demonstrating that the T6S defect observed in the mutants was due to a specific secretion defect and not due to impaired FPI expression (S6 Fig).

Taken together, these results demonstrate that the NBDs are very important for the virulence for *F. novicida*.

## The N-terminal of *F. novicida* ClpB is crucial for T6S and bacterial virulence

Based on the aforementioned findings, it is evident that ClpB of *F. novicida* has a unique mode of action and is not only a classical chaperone, but also critical for efficient T6S and bacterial virulence. To further delineate its role, we addressed whether it serves as a functional homolog of ClpV displaying unfoldase activity. For this activity, an α-helix (α0) of the N-terminal of *V. cholerae* ClpV binds to the N-terminal of VipB [39], but this α-helix is missing in *F. tularensis* ClpB (Fig 5A) [39]. Our analysis demonstrated that the N-terminal 56 amino acids of VipB demonstrate no similarity to any region of IglB (Fig 5B). Moreover, the N-terminus of IglB contains no α-helical regions similar to those predicted for VipB (Fig 5B); however, despite a very low overall sequence identity (~35%), the structural topology of IglB and VipB are similar (Fig 5C).

To address whether the N-terminal of the *F. novicida* ClpB plays a role for T6S, an N-terminal-truncated form of *F. novicida* ClpB (lacking amino acids 1–156) (Fig 6A), designated

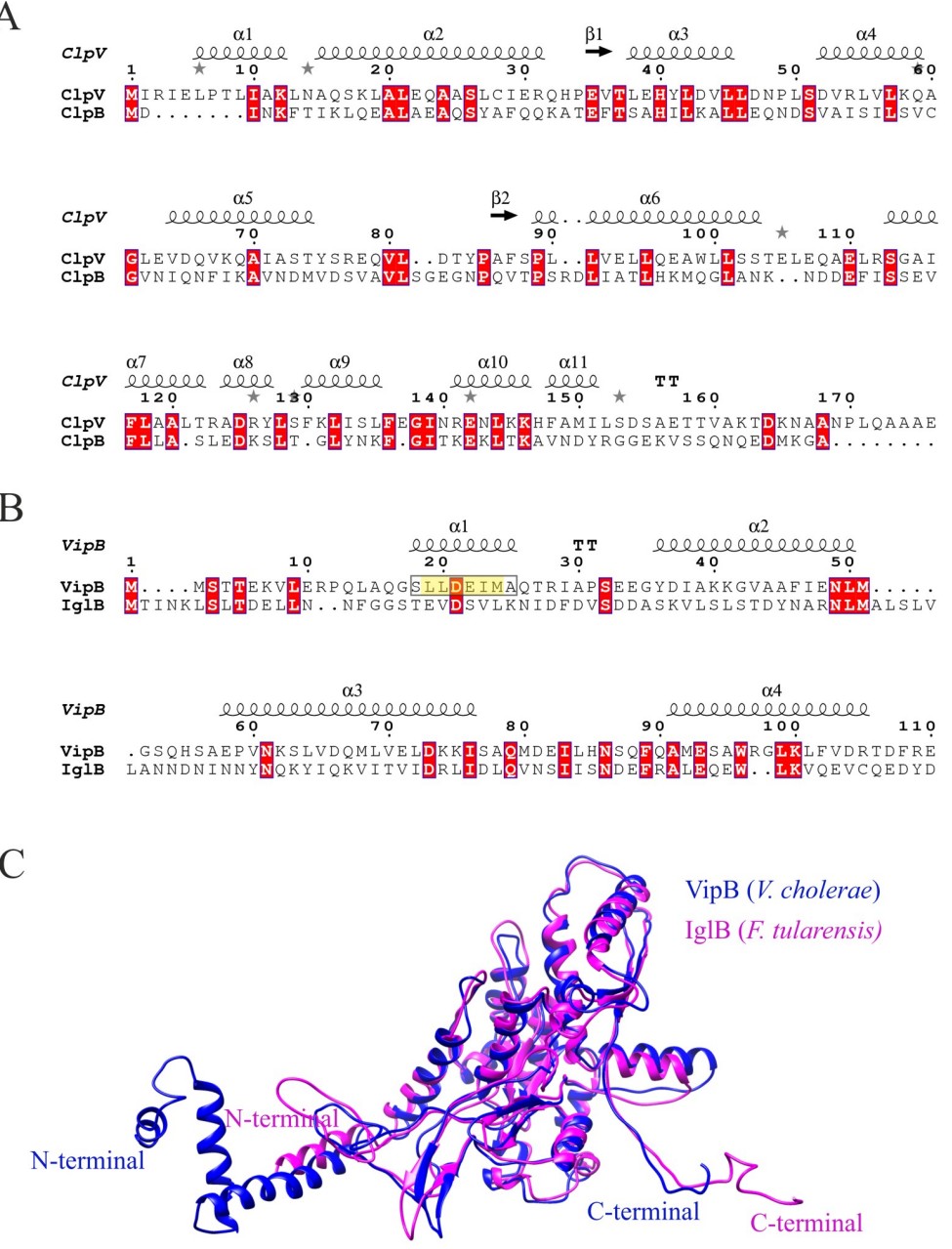

**Fig 5. Sequence alignment and structural comparison of the N-terminals of *V. cholerae* ClpV and VipB with *F. tularensis* ClpB and IglB.** (A). ClpV and ClpB sequences were retrieved from NCBI (https://www.ncbi.nlm.nih.gov/) and sequence alignments were performed using MAFFT (https://mafft.cbrc.jp/alignment/server/), and the corresponding image was generated using the web server ESPript 3 (http://espript.ibcp.fr). The first 174 amino acids (156 for *Francisella* ClpB) of the N-terminal domain of the ClpV-ClpB alignment is shown. Secondary structures as predicted for *V. cholerae* ClpV are displayed above the alignment. (B). The VipB and IglB sequences were retrieved from NCBI (https://www.ncbi.nlm.nih.gov/), and essentially the same procedure for alignment was performed as aforementioned. The first 110 amino acids, including the N-terminal domain (1–95 aa), of the VipB-IglB alignment is shown. Secondary structure elements as predicted for *V. cholerae* VipB are displayed above the alignment. The VipB α-helix known to interact with the ClpV N-terminal is boxed and highlighted in yellow. Conserved consensus sequence residues that contribute to the interaction with the ClpV-N-terminal are boxed and colored in yellow in helix 1. Identical amino acids are highlighted with red color. (C). Ribbon view of the *F. tularensis* IglB (IglB in pink; PDB: 3j9o) superimposed on the *V. cholerae* VipB (VipB in blue; PDB: 5mxn).

ΔN*clpB*, was generated and expressed *in trans* in the Δ*clpB* background. This truncation did not affect the overall conformation, thermal stability or the oligomeric status of the ClpB protein as confirmed by circular dichroism (CD) (S5A and S5C Fig), and gel filtration analysis (S2 Fig).

When tested for its effect on heat shock survival, substrate secretion and virulence in mice, Δ*clpB*/ΔN*clpB*$^+$ showed significantly better heat shock survival than did Δ*clpB* ($P < 0.05$; Fig 6B), however, still only 55% of the wild-type (Fig 6B), indicating that the N-terminal of ClpB of *F. novicida* is important, but not essential for heat shock survival.

To further investigate the role, the N-terminal of *F. novicida* ClpB was replaced with the N-terminal of *E. coli* ClpB, resulting in N*clpB*$_{EC}$. The mutant expressing the chimera demonstrated the same degree of heat shock survival as did U112 (Fig 6B), despite that the two N-terminals demonstrate only 36% identity (S7 Fig). In the KCl-induced substrate secretion assay, the level secreted by Δ*clpB*/ΔN*clpB*$^+$ was similar to the level of Δ*clpB*; whereas the level secreted by Δ*clpB*/N*clpB*$_{EC}$$^+$ was high and not significantly different from U112 (Fig 6C). When the intracellular replication of Δ*clpB*/ΔN*clpB*$^+$ and Δ*clpB*/N*clpB*$_{EC}$$^+$ was investigated, no significant difference were observed, and levels were similar to U112 (Fig 6D).

We next analyzed if the N-terminal truncation of ClpB affects the virulence in mice. When tested, the virulence of Δ*clpB*/N*clpB*$_{EC}$$^+$ in the mouse model was very similar to that of U112 and dramatically different compared to Δ*clpB*/ΔN*clpB*$^+$. The latter showed essentially no replication and the bacterial numbers were under the detection limit (Fig 6E).

In conclusion, the N-terminal of ClpB plays a vital role in substrate secretion and virulence, but less so for heat shock survival and none for intracellular replication. The ClpB chimera expressing the *E. coli* N-terminal demonstrates a wild-type phenotype with regard to all investigated properties.

## The ClpB function for heat shock survival, T6S and bacterial virulence is not species-specific

To address whether other parts of *clpB*, besides the N-terminal, have species-specific functions, we investigated substrate secretion and virulence in mice of Δ*clpB* expressing another bacterial member of the superfamily of ATPases, *E. coli clpB* (*clpB*$_{EC}$) *in trans*. Δ*clpB*/*clpB*$_{EC}$$^+$ fully complemented the heat shock survival (Fig 7A), substrate secretion (Fig 7B), and virulence in mice (Fig 7C), indicating that *E. coli* ClpB is a functional homolog of *F. novicida* ClpB (Fig 7A–7C). Of note, *clpB* of *F. novicida* demonstrates 64% sequence identity with *clpB*$_{EC}$ (S7 Fig). In conclusion, ClpB$_{EC}$ fully complemented the function of *F. novicida* ClpB, indicating that despite limited sequence similarity, there is extensive functional conservation among members of bacterial ATPases.

## Discussion

To date, little is known about the structure and biological role of molecular chaperones from *Francisella* spp., including ClpB. It has been demonstrated that the *Francisella* ClpB is essential for the pathogen's survival under stress conditions and also during infection of the host [5, 40, 41]. Moreover, we previously observed that *clpB* mutants of the clinically important subspecies *holarctica* and *tularensis* were defective for T6S [5] and we hypothesized that this impairment is the reason for the very marked attenuation of the *clpB* mutants in the mouse model.

There has been recent progress elucidating the structure of the atypical T6SS of *Francisella*. It has been demonstrated that it forms a sheath with a mesh-like architecture, comprising IglA/IglB [7], and ClpB was found to co-localize with the contracted sheath and also being necessary for the disassembly [6], as has been described for ClpV of prototypical T6SS. Thus, ClpB may be necessary for the normal function of T6SS, which requires disassembly.

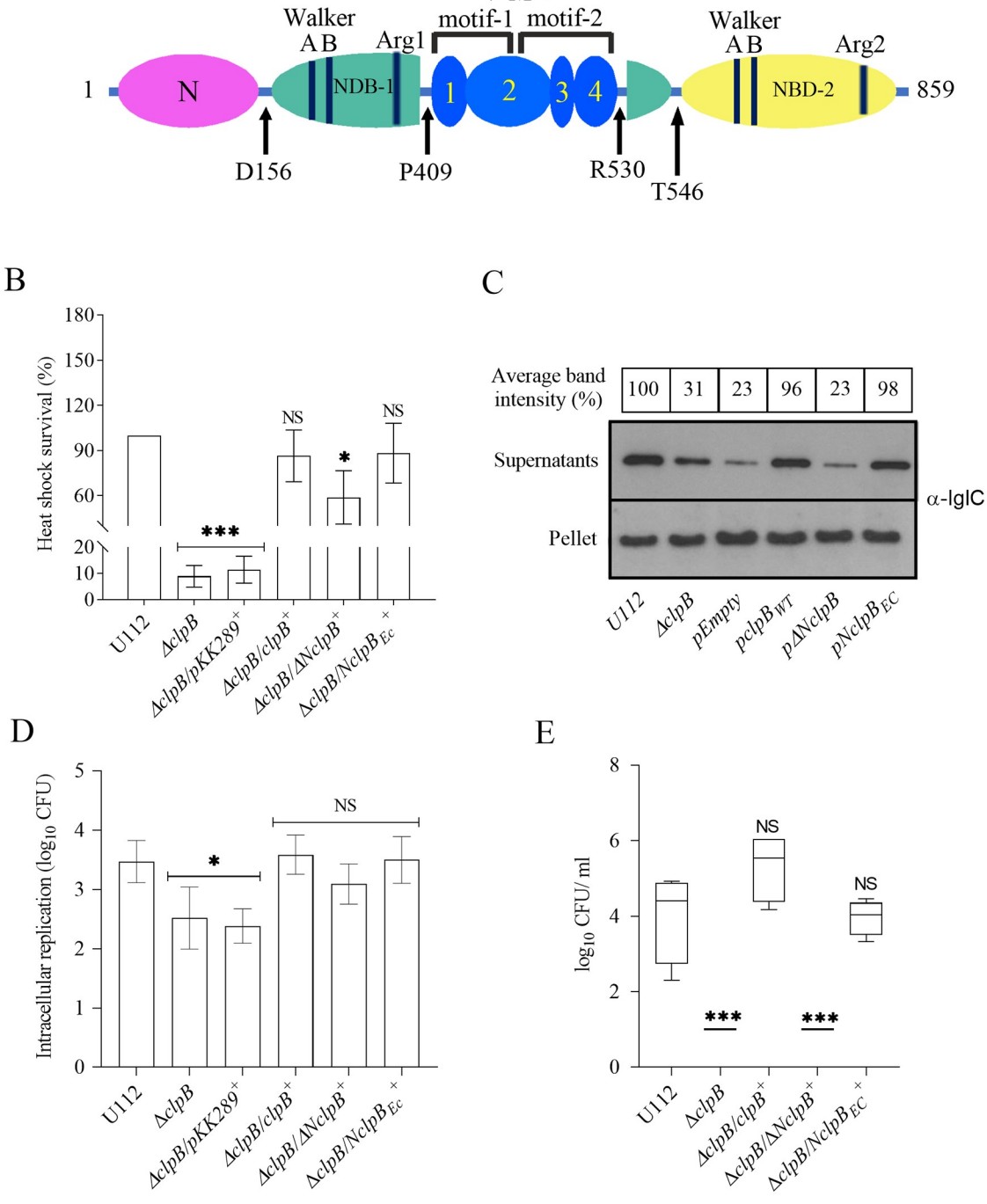

**Fig 6. The role of the N-terminal for the ability of *F. novicida* ClpB to support T6S and virulence in mice.** (A) Domain organization of *F. novicida* ClpB. The protein consists of an N-terminal (N) domain (magenta), two NDB domains (NDB-1, turquoise and NBD-2, yellow), and an inserted middle (M) domain (blue). The M domain contains four alpha-helices that are numbered accordingly. At the domain boundaries, the amino acid positions are indicated. (B) Heat shock survival of indicated strains. Bacteria were exposed to 50˚C for 30 min and the mean ± SD CFU are indicated. The wild-type strain U112 did not exhibit any significant killing during the treatment, and was set as 100%. A significant difference in the bacterial numbers of mutant strains *vs.* U112 is indicated as follows: *** $P < 0.001$; * $P < 0.05$; NS (not significant) $P > 0.05$. (C) Analysis of T6S by bacterial strains. Indicated strains were grown at 37˚C to an OD of 1.5 in TSB medium supplemented with 5% KCl. Precipitated supernatants or pellets of the same strain were separated by SDS-PAGE and analyzed using Western blot analysis using anti-IglC antiserum. At least three independent experiments were performed and a representative image is shown. The signal intensity of each band was

measured as described in Fig 3B and the percentage of the band-intensity *vs* U112 (set as 100%) is presented. (D) Indicated strains were used to infect J774A.1 cells. Infected cells were lysed at 0 h and 24 h and the number of CFU were determined. The net growth mean values ± SEM of at least three independent experiments are shown. A significant difference in the bacterial numbers of mutant strains *vs.* U112 is indicated as follows: * $P < 0.05$; NS (not significant) $P > 0.05$. (E) After subcutaneous inoculation with $1 \times 10^3$ CFU of the indicated *F. novicida* strains, mice were sacrificed on day 3, and bacterial burdens ($\log_{10}$ CFU/ml) in liver were determined. The mean ± SEM for six mice per group is indicated. A significant difference in the bacterial numbers of mutant strains *vs.* U112 is indicated as follows: *** $P < 0.001$; NS (not significant) $P > 0.05$.

In the present study, we demonstrate that, as expected, and in agreement with previous data [6, 40], ClpB displays an essential role for the heat shock survival of *Francisella*. Moreover, we demonstrate the essential roles of the Walker A, Walker B, and Arginine finger motifs, since mutations of a critical amino acid in any of the two copies of each of the motifs resulted in phenotypes essentially indistinguishable from that of the deletion mutant with regard to all investigated ClpB-dependent functions; heat shock response, T6S, intracellular replication, and virulence. The findings also demonstrate that both copies of each motif are important, since the lack of one copy cannot be fully compensated by the presence of the other. We observed that four of the mutants did not adapt the normal hexameric conformation and, likely, this explains their null phenotypes. In contrast, all other mutants showed no structural or conformational defects, thus, the logical explanation for their functional defects must be found in the mutated residue and not within the overall structure of the protein. The Walker B double mutant exhibited a null phenotype similar to the Δ*clpB* mutant with regard to T6S and virulence, likely a consequence of the abolished ATPase activity that was observed. In agreement, findings in *V. cholerae* have demonstrated that an ATP-driven remodeling activity of ClpV is needed for efficient T6S [42]. However, several of the NBD single mutants displayed an ATPase activity as high, or higher than ClpB, while all other tested phenotypes suggested that they were similar to Δ*clpB*. In this regard, it has been observed that a threading activity by ClpV is essential for T6S in *V. cholerae*. Both NBD domains are required for the threading

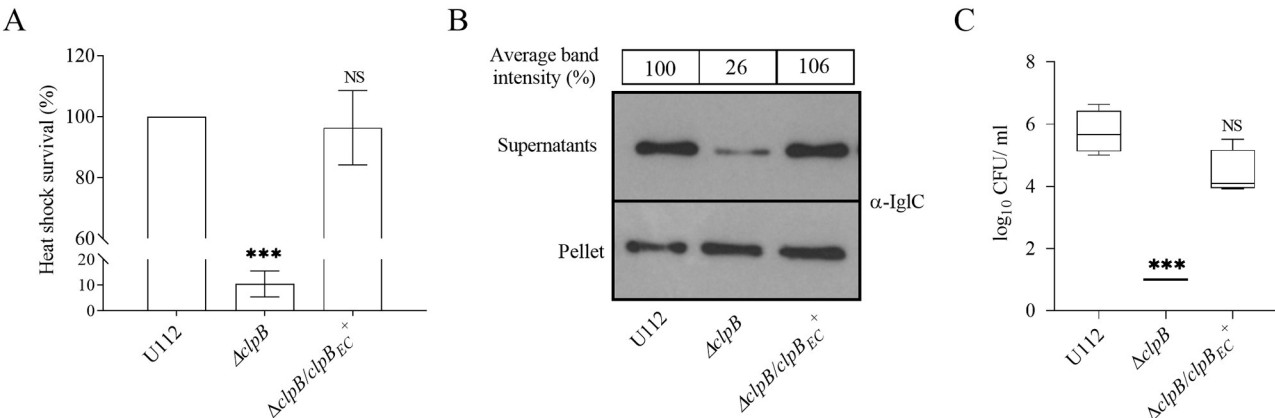

**Fig 7. *E. coli* ClpB phenotypically complements *F. novicida* ClpB.** (A) Heat shock survival of indicated strains. Bacteria were exposed to 50˚C for 30 min and the mean ± SD CFU is shown. The wild-type strain U112 did not exhibit any significant killing during the treatment, and the value was set as 100%. A significant difference in the bacterial numbers of mutant strains *vs.* U112 is indicated as follows: *** $P < 0.001$; NS (not significant) $P > 0.05$. (B) Analysis of T6S by bacterial strains. Indicated strains were grown at 37˚C to an OD of 1.5 in TSB medium supplemented with 5% KCl. Precipitated supernatants, or pellets of the same strain were separated by SDS-PAGE and analyzed using Western blot analysis with an anti-IglC antiserum. At least three independent experiments were performed and a representative image is shown. The signal intensity of each band was measured as described in Fig 3B and the percentage of the band-intensity *vs* U112 (set as 100%) is presented. (C) After subcutaneous inoculation with $1 \times 10^3$ CFU of the indicated *F. novicida* strains, mice were sacrificed on day 3, and bacterial burdens ($\log_{10}$ CFU/ml) in liver were determined. The mean ± SEM for six mice per group is indicated. A significant difference in the bacterial numbers of mutant strains *vs.* U112 is indicated as follows: *** $P < 0.001$; NS (not significant) $P > 0.05$.

activity of *E. coli* ClpB, since they work in alternating cycles, together enabling threading via a processive, rope-climbing mechanism [43]. Such threading, mediated by ClpV, modulates the conformation of the VipA-VipB complex, which is a prerequisite for effective T6SS assembly [42]. Based on these arguments, we hypothesize that the low levels of T6S observed in single *F. novicida* NBD mutants with retained or high ATPase activity are due to inefficient threading.

Importantly, mutagenesis of DnaK-interacting amino acids within ClpB demonstrated that an abrogation of the ClpB-DnaK interaction did not affect T6S, thus, this interaction is dispensable for the T6S. The prominent ATPase activity exhibited by some M-domain mutants was not unexpected, since certain *E. coli* variants thereof have demonstrated the same kind of enhanced ATPase activity, supporting the conclusion that the M-domain negatively regulates the substrate-stimulated ATPase activity of ClpB [25, 33, 34]. Notably, despite their much enhanced ATPase activity, the affected M-domain mutants of *Francisella* ClpB still exhibited the same level of T6S and virulence as did the wild-type strain. Thus, the ATPase activity of ClpB appears to be sufficient for efficient T6S. Interestingly, the highly thermosensitive Y503 mutant showed intact virulence, strongly implying that T6S, but not adaptation to heat shock, is the critical function of ClpB required for the *F. novicida* virulence. Thereby, ClpB plays a unique dual role in *F. novicida*, not only related to its chaperone function. Since the ClpB protein is >99% conserved between species, we expect that our findings are fully applicable to *F. tularensis*.

The degree of intracellular replication showed close correlation to the degree of survival during heat shock, *i.e.*, the mutants most susceptible to heat shock showed the least growth. Importantly, however, even the latter mutants showed significant growth, thus, they demonstrated a phenotype very distinct from an FPI mutant lacking a functional T6SS, which demonstrates no intracellular growth [3, 35]. Our findings regarding the U112 Δ*clpB* mutant are in agreement with the previous study by Woolard that demonstrated significant intracellular replication [41], whereas, in contrast, the study by Brodmann et al did not demonstrate any replication of a Δ*clpB* mutant [6]. Notably, we observed that the deletion of the ClpB N-terminal resulted in abrogation of virulence in the mouse model, but still intact intra-macrophage replication. Although this may appear paradoxical, the finding is not unprecedented, since a similar phenotype is displayed by a few mutants of T6SS components, *e.g.*, Δ*iglG* and Δ*iglI* [44]. Whereas intra-macrophage replication usually closely correlates with virulence in animal models, the findings implicate that animal virulence is more complex and may, *e.g.*, depend also on intracellular replication in other cell types than macrophages, as demonstrated by the phenotype of Δ*clpB*/ΔN*clpB*$^+$.

Our *in silico* modeling demonstrated the extensive conformational conservation of ClpB proteins, since only minor differences, with the exception of the N-terminal, were noted between the *Francisella* protein and the prototypical ones of *E. coli* and *T. thermophilus*. The finding that the chimeric ClpB protein demonstrated the same degree of heat shock survival as U112, despite that the two N-terminals demonstrate only 36% identity, was not totally unexpected, since the N-terminal is not responsible for the species-specificity of the thermotolerance [45]. Despite these differences, we observed that the *E. coli* homolog functionally complemented the critical role of the *F. novicida* ClpB also for T6S and virulence. This demonstrates that the unique role of ClpB in *F. novicida* may be due to the lack of a ClpV homolog. The findings were somewhat surprising as the previously identified ClpV-interacting motif, an α-helical region at the N-terminal of *V. cholerae* VipB is missing in IglB, the VipB homolog [39]. Moreover, our bioinformatic analysis indicated that the N-terminal of IglB has no similar α-helical region and the N-terminal 56 amino acids of VipB demonstrate no similarity to any region of IglB. A direct structural comparison between N-terminals of IglB and VipB could not be performed, since some of the predicted N-terminal helices were not visualized in the cryo-EM structures of the homologs VipB and

TssC1 of *P. aeruginosa* [46, 47]. It was proposed that these parts are buried in the extended protein, whereas, once the sheath contracts, the N-terminal ClpV-binding domain is exposed, thereby facilitating sheath disassembly [47]. Although no direct evidence of a ClpB-IglB interaction is available and we failed to purify soluble IglA-B to provide such evidence, co-localization of IglA and ClpB has been observed by live-cell microscopy [6, 48], similar to that observed for ClpV and VipB [13, 14]. Thus, if there is an interaction, then this must be distinct from that of ClpV-VipB, since there are no regions in *Francisella* ClpB and IglB with similarity to the interacting regions of ClpV and VipB

Collectively, our data demonstrate a critical role of ClpB for T6S and virulence of *F. novicida* and that the Walker and Arginine finger motifs are essential for T6S, whereas the ClpB-DnaK is exclusively related to the handling of stress stimuli, such as heat shock, but not to virulence. Moreover, we observed that the N-terminal of *F. novicida* was not essential for heat shock survival, but critical for T6SS and bacterial virulence. Thus, there is structural dissociation between the critical roles of ClpB for *F. novicida*. The study provides essential information about the control and regulation of the T6SS of *Francisella* and possibly T6SS of other bacterial species that lack ClpV.

## Materials and methods

### Bacterial strains, media and culture conditions

The bacterial strains and plasmids used in this study are listed in S1 Table. *Escherichia coli* strains were grown either in Luria Bertani broth (LB) or on Luria agar plates (LA) at 37 ˚C. *F. novicida* strains were cultured either in Tryptic soy broth (TSB) supplemented with 0.1% cysteine (w/v) and 0.1% glucose (w/v), or in Chamberlain's defined medium (CDM) [49] at 37 ˚C with shaking at 250 rpm, or on modified GC-agar at 37 ˚C, 5% $CO_2$. When required, kanamycin (50 μg/ml for *E. coli* or 10 μg/ml for *F. novicida*), carbenicillin (100 μg/ml), polymyxin B (50 μg/ml) or chloramphenicol (34 μg/ml for *E. coli*, 8 μg/ml for *F. novicida*) was added to the medium. For the substrate secretion assay, *F. novicida* and mutants thereof were cultured in TSB medium with or without 5% KCl.

### Ethics statement

Mice were housed and handled in agreement with good animal practice as defined by EU directive EU 2010/63 and ETS 123 and the Swedish regulations Animal Welfare Ordinance, the Animal Welfare Act, and SJVFS 2012:26. The animal experiments were performed in accordance with the Swedish animal protection law and were approved by the local Ethical Committee on Laboratory Animals, Umeå, Sweden, approval no. A67-14.

### Mouse infection

Mice were obtained from Charles River Laboratories, Sulzfeld, Germany and housed in the animal facility of the Umeå Centre for Comparative Biology under SPF conditions according to FELASA recommendations. For determination of the virulence strains, C57BL/6 female mice (n = 6) were infected subcutaneously with approximately $1 \times 10^3$ CFU for each *F. novicida* strain. Mice were examined twice daily for signs of severe infection and euthanized by $CO_2$ asphyxiation as soon as they displayed signs of irreversible morbidity. In our experience, such mice were at most 24 h from death, and the time to death of these animals was estimated on the basis of this premise. At days 3 and 5, mice infected with U112, Δ*clpB*, or ClpB mutant variants were killed and serial dilutions of the homogenized organs were plated for

determination of viable counts. A two-sided t-test with equal variance was used to determine whether the growth of the mutants differed significantly from the wild-type strain.

## Sequence conservation analysis, comparative protein modeling and prediction of protein stability

For comprehensive evolutionary sequence conservation analysis, the ConSurf Program [50] was used. The primary protein sequence of *F. novicida* U112 was considered as a query to identify homologues sequences of ClpB in the protein database UNIREF-90 (https://www.uniprot.org/) using the HMMER (hidden Markov models) homolog search algorithm [51] with an E-value cutoff 0.0001. Best top 500 hits were used to build the Multiple Sequence Alignment (MSA). The MSA was performed using the MAFFT program [52]. All other parameters were kept at default values for calculation of conservation scores.

To generate the model of *F. novicida* ClpB and DnaK, the comparative protein modeling was performed for both proteins using the template structures PDB ID: 1QVR (for ClpB) [30] and 2KHO (for DnaK) [31] via SWISS-MODEL server (https://swissmodel.expasy.org) [53]. The modeled protein structures were energetically minimized by 100 steps of steepest descent method following 50 steps of conjugate gradient method using the minimize structure module of the UCSF Chimera software [54]. The minimized modeled structures were used for protein-protein (ClpB-DnaK) docking studies.

The Site Directed Mutator (SDM) method and Multi agent stability prediction (MAE-STRO) were used to predict the stability change of mutated proteins [55]. The SDM method analyzes the variation of amino acid replacements occurring at specific structural environment that are tolerated within the family of homologous proteins of known 3-D structures and convert them into substitution probability tables. These tables are used as a quantitative measure for predicting the protein stability upon mutation. MAESTRO is structure-based and implements a multi-agent machine learning system. It provides high throughput scanning for multi-point mutations where sites and types of mutation can be comprehensively controlled.

The *F. novicida* and *F. tularensis* ClpB amino-acid compositions are very similar, only 7 out of 859 amino acids differ and all of the differences are located in the N-terminal and C-terminal parts of the protein. There are no differences between the ClpB proteins of *F. tularensis* LVS and SCHU S4.

## Predictions of ClpB—DnaK protein-protein interactions

To explore the interaction mode of binding partner protein of ClpB, the HADDOCK2.2 (High Ambiguity Driven protein-protein DOCKing) program [56] was used. This program uses an information-driven flexible docking approach for the modeling of biomolecular complexes and protein-protein docking studies. Residues were defined based on the sequence conservation analysis as well considering those already reported in homologous proteins. The remaining parameters were kept default during the docking run. The best-docked conformation of ClpB-DnaK was selected based on key interactions and the docked score and subjected to molecular dynamics (MD) simulation studies to attain the optimum interactions. For the MD simulation, the AMBER 17.0 software package using ff14SB force fields was used [57]. Using the tleap module, all required parameters were employed, considering ionizable residues set at their default protonation states at a neutral pH value. The ClpB-DnaK complex was neutralized by adding 33 $Na^+$ ions and solvated in a truncated octahedron box of the TIP3P [58] water model with a margin distance of 10 Å. To account for long-range Coulombic interactions, the particle mesh Ewald method was used with a cut-off of 10.0 Å [59]. The SHAKE algorithm [60] was employed to restrain all atoms covalently bonded to hydrogen atoms,

allowing for an integration time step of 2 fs. Periodic boundary conditions were imposed to avoid edge effects. The box was minimized by 750 steps of the steepest descent method following 500 steps of the conjugate gradient method, while restraining the protein using a force constant of 2 kcal/mol $Å^2$. The system was gradually heated from 0 to 300 K over a period of 50 ps and maintained at 300 K with a force constant of 2 kcal/mol Å2. The system was equilibrated for 1.0 ns. Finally, a production run for 100 ns was performed using NPT ensemble at room temperature with 1.0 atm pressure. Coordinate trajectories were recorded every 20 ps for further analysis using the UCSF Chimera [54] and VMD programs [61].

## Cloning and purification of ClpB or ClpB variants, DnaJ, DnaK and GrpE of *F. novicida*

The full length *clpB*, *dnaJ*, *dnaK*, and *grpE* genes of *F. novcida* were codon-optimized and synthesized by GenScript Corp (https://www.genscript.com/) and expressed in *E. coli*. Genes were sub-cloned between the *NcoI* and *XhoI* sites of pET His-1a expression vector (a modified pET vector, obtained from the protein expression and purification facility, Umeå University, Sweden) containing a 6-histidine tag followed by a tobacco etch virus (TEV) protease cleavage site. The substitution variants of *clpB* were introduced using the Quikchange XL-II site directed mutagenesis kit (Stratagene) and verified by DNA sequencing.

   To express the recombinant proteins, *E. coli* BL21(DE3)LysS (Novagen) cells expressing ClpB or ClpB variants, DnaJ, DnaK, or GrpE were grown in LB broth containing 50 µg/mL kanamycin. Cells were grown to $OD_{600}$ ~ 0.8 and expression was induced by addition of 1 mM Isopropyl β-D-1-thiogalactopyranoside (IPTG) and grown further overnight at 25 ˚C. Next day, bacteria were harvested at 4˚ C and the proteins were purified on Ni-NTA resin (Qiagen), followed by cleavage of the tagged N-terminal 6-histidine using TEV protease and a HiPrep DEAE FF 16/10 column. The cleaved proteins were concentrated on an Amicon Ultra-15 30K molecular weight cutoff (MWCO) filter (Millipore) and further purified on a HiLoad 16/60 Superdex 200 pg gel filtration column (GE Healthcare), equilibrated with 50 mM Tris pH 7.5, 150 mM KCl, 20 mM MgCl2. For DnaK and GrpE, addition of guanidine-HCl (6 M) with heating (80˚C) was required to denature DnaK or GrpE to allow re-protonation of all amide groups in the perdeuterated protein and/or complete removal of the bound nucleotides or *E. coli* contaminants. The proteins were purified on an Ni-NTA resin, refolded on the column and followed by cleavage of the purification tag using TEV protease and a HiPrep DEAE FF 16/10 column. DnaK was further purified on a HiLoad 16/60 Superdex 200 pg gel filtration column (GE Healthcare), equilibrated with 50 mM Tris pH 7.5, 150 mM KCl, 20 mM MgCl2. The purity of all proteins was confirmed by SDS-PAGE.

## Gel-filtration analysis

ClpB or ClpB variants (500 µL of ~5 mg/ml protein) were incubated in running buffer (50 mM Tris, pH 7.5; 20 mM $MgCl_2$; 150 mM KCl, and 5% (v/v) glycerol) in the presence of 2 mM ATP for 5 min at 25˚C, followed by injection into the high pressure liquid chromatography system (GE Healthcare) connected to a Superdex 200 Increase 10/300 GL (GE Healthcare). Chromatographic steps were performed with a flow rate of 0.5 ml/min. Molecular size standards were purchased from Sigma-Aldrich, St. Louis, MO, USA.

## Circular dichroism spectroscopy

Far-UV Circular Dichroism (CD) spectra were recorded between 200–250 nm at 25˚C using a Jasco J-720 Spectropolarimeter equipped with a Peltier temperature controller (Japan). The protein concentration was 10 µM in 10 mM NaPi, 30 mM NaCl, at pH 7.5. The spectra were

recorded using a 0.1 cm quartz cuvette, a bandwidth of 2 nm with subtracted background, and data were averaged based on five repeated scans. The experimental conditions used for temperature unfolding were the same, but only used single scan acquisition. To assess the thermal stability of ClpB or the ClpB variants, the far-UV CD signal at 220 nm was recorded between 20 and 75°C using a scan rate of 0.5 °C / min. The transition mid-point temperature ($T_m$) was calculated by fitting the sigmoidal Boltzman curve to the ellipticity data using the program OriginPro 9.1 (OriginLab Corp., USA, www.originlab.com). For monitoring reversible thermal unfolding transitions, the temperature was increased stepwise (1°C/min, 10–85 °C). Temperature-induced changes of the CD signal at 220 nm were analyzed by using a two-state thermodynamic model.

The midpoint of thermal unfolding ($T_m$) was obtained by fitting CD data to equations below,

$$CD_{norm}(T) = \frac{S_f + \alpha T + K_{obs}(S_u + bT)}{1 + K_{obs}}$$

where

$$K_{obs}(T) = exp\left(\frac{\Delta H_m}{R}\left(\frac{1}{T_m} - \frac{1}{T}\right)\right)$$

$CD_{norm}(T)$ is the normalized CD signal; $S_f$, $S_u$, a, and b are the CD signals for folded and unfolded conditions and the slopes for the folded and unfolded baselines, respectively. $\Delta H_m$ corresponds to the enthalpy value at $T_m$.

## Disaggregation activity assays

ClpB disaggregation activities were performed by following the disaggregation of heat-aggregated Malate Dehydrogenase (MDH) (0.5 µM, 30 min at 47°C) and urea-denatured recombinant firefly luciferase (0.2 µM, 30 min at 30°C) as described elsewhere [20, 34]. Chaperones of *F. novicida* were used at the following concentrations: 1 µM ClpB (wild-type or its variants), 1 µM DnaK, 0.2 µM DnaJ, and 0.1 µM GrpE. Disaggregation reactions were carried out in a reaction buffer (50 mM Tris pH 7.5, 150 mM KCl, 20 mM MgCl$_2$, 2 mM DTT) containing an ATP-regenerating system (2 mM ATP, 3 mM phosphoenolpyruvate, 20 ng/µl pyruvate kinase). MDH disaggregation was monitored by turbidity measurement at an excitation and emission wavelength of 600 nm (Tecan Infinite F200). Considering the initial MDH turbidity as 100%, data were calculated compared to the denatured MDH and shown in percentage. For luminescence, reactions were incubated at 23°C for 60 min, aliquots (5 µL) were removed at the times indicated and luciferase activity was determined by adding 50 µM luciferin (Promega) and measuring light output in a Tecan Infinite F200. Reactivation of luciferase was determined compared to a non-denatured luciferase control.

## ATPase activity assay

The ClpB ATPase activity was determined in the absence or presence of 10 mM casein in reaction buffer (50 mM Tris pH 7.5, 150 mM KCl, 20 mM MgCl$_2$, 2 mM DTT) using a NADH-coupled colorimetric assay. The decrease of NADH absorption was at 340 nm using a Tecan Infinite F200 plate reader. Typically, 0.5 µM of ClpB or ClpB mutants were used in the reactions, except for the ClpB M-domain mutants Q502A or Y503A (0.10 µM in the presence of casein) and E500A or K508A (0.20 µM in the presence of casein). The four latter were used at

lower concentrations since their ATPase activity was much higher compared to the wild-type. The ATPase activity was calculated based on the linear decrease of NADH absorbance.

## Construction of *clpB* mutants and *in cis* complementation

The *F. novicida* U112 Δ*clpB* strain was generated by allelic replacement essentially as described previously [62]. Flanking regions upstream and downstream of the gene were amplified by PCR and then a second overlapping PCR were performed using purified fragments of the first PCR as a template. The overlap PCR fragment was then cloned to suicide vector pDMK3 and the resulting plasmid (pALA012) was first introduced into *E. coli S17-1λpir* and then transferred to *F. novicida* by conjugation. Clones with plasmids integrated into the *clpB* gene of the *F. novicida* chromosome by a single recombination event were selected on plates containing kanamycin and polymyxin B and verified by PCR. These clones were then subjected to sucrose selection on Mueller-Hinton plates with 10% sucrose. This procedure selected for a second cross-over event in which the integrated plasmid, encoding *sacB*, was excised from the chromosome. Kanamycin-sensitive, sucrose-resistant clones were examined by PCR and clones containing the deletion of the *clpB* gene confirmed. The *in cis* complementation of the *clpB* mutant was based on the essentially same procedures; however, the upstream and downstream region was amplified together with the wild-type *clpB* gene (pALA013). To generate constructs encoding specific mutations within ClpB, plasmid pALA013 was used as template using the QuikChange II XL Site-Directed Mutagenesis Kit (Agilent technologies, Stockholm, Sweden AB). By this approach, constructs encoding single substitution mutations within Walker A (K212A), Walker B (E279A) and Arginine finger motifs (R332A) of the first nucleotide-binding domain (NBD-1) of *clpB* were generated and designated *WA1*, *WB1* and *Arg1*, respectively. Similarly, single substitution mutations within Walker A (K613A), Walker B (E680A) or Arginine finger (R757A) of NBD-2 of *clpB* were generated and designated *WA2*, *WB2*, and *Arg2*. The Walker A mutation affects a lysine residue and this impairs nucleotide binding (S4 Fig) and the Walker B mutation affects a glutamate residue and this prevents the hydrolysis of ATP (S4 Fig) [43, 63]. Mutants with substitutions in the arginine finger result in the interruption of ClpB oligomerization, ATP binding and hydrolysis [64]. The amino acid residues of each NBD were identified based on the conservation analysis and multiple sequence alignment (S1 and S4 Figs). Constructs encoding double substitution mutants within both copies of Walker A (K212A/K613A), Walker B (E279A/E680A) and Arginine finger motifs (R332A/R757A) were also generated and designated *WA1-2*, *WB1-2* and *Arg1-2*, respectively. The M-motif substitution mutants of ClpB were generated using the same aforementioned protocol.

For complementation *in trans*, plasmid pKK289Km-*clpB*, carrying the *clpB* wild-type gene under the control of the LVS *groES* promoter, was used [65]. Using a strategy similar to the aforementioned site-directed mutagenesis, the middle domain variants of the *clpB* were generated. To generate the chimera containing the N-terminal of *E. coli* to the *F. novicida* *clpB*, a full-length gene containing the N-terminal of *E. coli* (1–156 aa) fused with the *F. novicida clpB* (157–857 aa) was synthesized from the GenScript Corp and cloned directly into the pKK289Km vector. Full-length *clpB* from *E. coli* (str. K-12 substr MG1655) was PCR amplified and cloned directly to the pKK289Km vector. All constructs were sequenced and PCR was used to verify that the anticipated genetic event had occurred followed by RT-PCR to ensure that *clpB* and mutant variants thereof were transcribed. All the plasmids used for *in trans* complementation were then introduced into the Δ*clpB* mutant by cryotransformation. Primers used are listed in the S2 Table.

## Cultivation and infection of macrophages

The murine macrophage-like cell line J774A.1 (American Type Culture Collection, Manassas, Va), or bone marrow-derived macrophages (BMDMs) were used in the cell infection assays. J774A.1 macrophages were cultured and maintained in DMEM (GIBCO BRL, Grand Island, NY, USA) with 10% heat-inactivated FBS (GIBCO) at 37˚C with 5% $CO_2$. BMDMs were isolated by flushing bone marrow cells from the femurs and tibias of C57BL/6 mice as described previously [5]. The day before infection, cells were seeded in tissue culture plates, incubated overnight, and reconstituted with fresh culture medium at least 30 min prior to infection to recover. Plate-grown bacteria were suspended in PBS and kept on ice prior to infection. A multiplicity of infection (MOI) of 200 was used in all infection experiments.

## Bacterial viable counts

To determine the intra-macrophage growth of *F. novicida*, J774A.1 or BMDM cells were infected for 2 h, washed 3 times, and incubated in fresh medium supplemented with 5 μg/ml gentamicin for 30 min (corresponds to time zero). At indicated time points, the macrophage monolayers were lysed in 0.1% deoxycholate, serially diluted in PBS and plated on modified GC-agar base plates for determination of viable counts. A two-sided t-test with equal variance was used to determine whether the growth of the mutants differed significantly from the wild-type strain.

## Temperature susceptibility test

*F. novicida* were grown overnight on modified GC-agar plate, resuspended in pre-equilibrated Chamberlain medium at 37˚C to give an $OD_{600}$ of 0.5, and 0.25 ml of the culture was placed statically in a water bath at 50˚C for 30 min. The tubes were then serially diluted in sterile PBS and plated on modified GC agar plates to determine the number of viable bacteria. At indicated time points, aliquots were sampled, serially diluted in sterile PBS and plated to determine the number of viable bacteria.

## KCl-induced substrate secretion assays

*In vitro* KCl-induced substrate secretion assay was performed as reported recently with slight modifications [7]. Briefly, *F. novicida* and mutants thereof were grown on modified GC agar plates overnight. Next day, bacteria was suspended to $OD_{600}$ ~ 0.15 in fresh TSB medium (supplemented with 0.1% cysteine and 0.1% glucose) with or without 5% KCl and cultivated to an $OD_{600}$ ~ 1.5. Cultures were then centrifuged at $4000 \times g$ for 10 min at 4˚C and pellets harvested. The supernatants were filtered-sterilized using a 0.22 μm syringe filter and proteins precipitated by the TCA precipitation method. Protein fractions were separated by SDS-PAGE and analyzed using standard Western blot analysis and the Enhanced Chemiluminescence system (ECL) (Amersham Biosciences, Uppsala, Sweden). IglC was detected using an IglC antibody (BEI resources, Manassas, VA, USA), followed by an anti-mouse secondary antibody coupled to horseradish peroxidase (Santa Cruz Biotechnology, CA, USA).

## Western blot analysis

The TCA-precipitated supernatant fractions and the corresponding pellets were prepared in sample buffer and boiled before applied on a 12% sodium dodecyl sulfate (SDS)-polyacrylamide gel for separation. For Western blot analysis, the proteins were transferred onto a nitrocellulose membrane using a semi-dry blotter (Bio-Rad laboratories, CA, USA). To determine the protein level of different FPI proteins in bacterial lysates, essentially the same protocol was

used as reported earlier [66]. Blots were probed by either rabbit polyclonal antibodies against IglA, or mouse monoclonal antibodies against IglB (BEI Resources), IglC, rat polyclonal antibodies against IglE [67], rabbit polyclonal antibodies against IglH, VgrG, (Inbiolabs, Tallinn, Estonia), or PdpA, PdpB and PdpC (Agrisera, Vännäs, Sweden). Polyclonal IgY chicken antibodies, specific to IglD or FupA were used (Agrisera). Horseradish peroxidase (HRP) conjugated secondary antibodies goat anti-mouse (Santa Cruz Biotechnology, CA, USA), donkey anti-rabbit (GE Healthcare, UK), and rabbit anti-chicken IgY (Sigma-Aldrich, St. Louis, MO, USA) were used. For detection, the Enhanced Chemiluminescence system (ECL) was used.

## Statistical analysis

For the statistical analysis, GraphPad Prism 7 (GraphPad Software Inc., CA, USA) was used. Unpaired t-test with Welch's correction was used to compare values.

## Supporting information

**S1 Fig. Evolutionarily conserved functional and structurally important residues of *Francisella* ClpB and DnaK. (A)** For the conserved score calculation, the query sequence of ClpB (Accession No:AJJ47289) was used as input in the ConSurf program. The boxes represent the residues of the Walker motifs and the important residues of Arginine fingers. WA1; Walker A motif of nucleotide binding domain (NBD)-1; WA2; Walker A motif of NBD-2; WB1; Walker B motif of NBD-1; WB2; Walker B motif of NBD-2; Arg-1: Arginine finger residue of NBD-1; and Arg-2: Arginine finger residue of NBD-2. **(B)** For the conserved score calculation, the query sequence of DnaK (Accession No: APC95585) was used as input in the ConSurf program. Conservation scale is from 1 to 9 and the color codes for the conservation are depicted in the figure. **e**—An exposed residue according to the neural-network algorithm. **b**—A buried residue according to the neural-network algorithm. **f**—A predicted functional residue (highly conserved and exposed). **s**—A predicted structural residue (highly conserved and buried). (PDF)

**S2 Fig. Gel filtration analysis of ClpB or the ClpB variants.** Elution profile of ClpB, or the ClpB variants were determined in the presence of 2 mM ATP in running buffer as described in Materials and Methods. Elution profiles of four proteins in each sub-figures are placed together for better visibility. Molecular size standards used were Thyroglobulin (669 kDa), Apoferritin (443 kDa), Amylase (200 kDa), and Alcohol Dehydrogenase (150 kDa) and their positions are indicated. (PDF)

**S3 Fig. Analysis of the total levels of ClpB proteins from the wild type and its variants.** Whole cell lysate of wild type, Δ*clpB* or M-domain variants complemented *in trans* in Δ*clpB* were prepared, separated by SDS-PAGE and probed with ClpB antibody from *Synechocytis* PCC 6803 (slr1642, Agrisera) that cross reacts with the *Francisella* ClpB. Anti-IglB was used as a loading control. Vector indicates the empty vector (pKK289). Asterisks indicate non-specific bands. Assays were repeated at least twice and a representative blot is shown. (PDF)

**S4 Fig. Sequence alignment and conserved domains involved in ATP binding and ATP hydrolysis of *F. tularensis* ClpB.** ClpB sequences was retrieved from the NCBI server (https://www.ncbi.nlm.nih.gov/). Sequence alignments of *V. cholerae* (Accession: AKB07899.1), *F. novicida* U112 (Accession: AJI61375.1), *F. tularensis* LVS (Accession: CAJ78535.1), *F. tularensis* SCHU S4 (Accession: YP_170660.1), *E. coli* (Accession: BAA16476.1), *Mycobacterium tuberculosis* (Accession: LB17500.1), and *Pseudomonas aeruginosa* (Accession:

WP_058169183.1) were performed using MAFFT (https://mafft.cbrc.jp/alignment/server/) and the corresponding image generated using the server ESPript 3 (http://espript.ibcp.fr). Secondary structure elements based on the *V. cholerae* ClpB crystal structure are displayed above the alignment. Start and end of the N-terminal domain is marked with brown arrows. Conserved Walker A (206-G*X*4GKT-213 and 605-G*X*4GKT-612) and Walker B (276-Hy2DE-279 and 675-Hy2DE-678) motifs, where *X* = any amino (*aa*) and Hy = hydrophobic amino acids are indicated with blue boxes. Conserved Arg residues (R) present in both nucleotide binding domains (Arg-332 and Arg-756) are proposed to serve as Arg fingers and indicated with a blue box, The conserved residue of each of these motifs (highlighted in yellow) was subjected to alanine mutagenesis.
(PDF)

**S5 Fig. Circular dichroism, thermal stability, and Western Immunoblot analysis of the ClpB proteins. (A)** Far-UV CD spectra of the purified wild type ClpB and the mutant variants recorded between 200 nm– 250 nm at 25˚C. The protein concentration was 10 μM in 10 mM NaPi, 30 mM NaCl, at pH 7.5. The CD signal was expressed as the mean molar residue ellipticity. **(B-C)** Temperature-induced changes in the CD signal at 220 nm of WT and ΔN ClpB were recorded between 20 and 75˚C using a scan rate of 0.5 ˚C / min. The solid line shows the fit of a two-state unfolding model. **(D)** Western immunoblot analysis of the total cell lysates from the indicated strains probed with ClpB antibody from *Synechocytis* PCC 6803 (slr1642, Agrisera) that cross reacts with the *Francisella* ClpB. IglB was used as loading control. Asterisks indicate non-specific bands. Assays were repeated at least twice and representative blots are shown.
(PDF)

**S6 Fig. Analysis of the total levels of the FPI proteins of the indicated *F. tularensis* strains.** Whole cell lysate of each strain of ClpB variants and wild type was prepared, separated by SDS-PAGE and probed with specific antibodies against indicated FPI proteins. U112: *F. novicida* wild type, Δ*clpB*: *clpB*-deleted strain, and Δ*fpi*: *Francisella* Pathogenicity Island (FPI)-deleted *F. novicida* strain. Details about the antibodies used are described in Materials and Methods. Asterisks indicate non-specific bands. Assays were repeated at least twice and representative blots are shown.
(PDF)

**S7 Fig. Sequence alignment of the N-terminal of the *E. coli* and *F. novicida* ClpB.** ClpB sequences of *E. coli* and *F. novicida* U112 were retrieved from NCBI (https://www.ncbi.nlm.nih.gov/), sequence alignments were performed using MAFFT (https://mafft.cbrc.jp/alignment/server/), and the corresponding image was generated using the web server ESPript 3 (http://espript.ibcp.fr). The first 180 of N-terminal domain (1–156 aa) of the *E. coli*-U112 ClpB alignment is shown. Secondary structure elements as predicted for *E. coli* ClpB are displayed above the alignment.
(PDF)

**S1 Table. Strains and plasmids used in this study.**
(DOCX)

**S2 Table. Oligonucleotides used in the study.**
(DOCX)

**S3 Table. Molecular Dynamic Simulation (MDS), 100ns.**
(DOCX)

**S4 Table. Predicted change in protein stability upon introduction of point mutations.**
(DOCX)

**S1 Video. Molecular Dynamic Simulation (MDS), 100ns.**
(AVI)

## Acknowledgments

Jeanette Bröms is gratefully acknowledged for the plasmids pKK289Km and valuable discussion and critical reading of the manuscript.

## Author Contributions

**Conceptualization:** Athar Alam, Anders Sjöstedt.

**Formal analysis:** Athar Alam, Jörgen Ådén.

**Funding acquisition:** Anders Sjöstedt.

**Investigation:** Athar Alam, Igor Golovliov, Eram Javed, Rajender Kumar, Jörgen Ådén.

**Project administration:** Anders Sjöstedt.

**Resources:** Anders Sjöstedt.

**Software:** Rajender Kumar.

**Supervision:** Anders Sjöstedt.

**Writing – original draft:** Athar Alam, Anders Sjöstedt.

**Writing – review & editing:** Anders Sjöstedt.

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
