## [Decision Letter · Decision Letter 0]

12 Dec 2019

Dear Dr. Sjostedt,

Thank you for submitting your manuscript "Dissociation between the critical role of ClpB of Francisella tularensis for the heat shock response and the DnaK interaction and its important role for efficient type VI secretion and bacterial virulence" (PPATHOGENS-D-19-01936) for review by PLOS Pathogens. Your manuscript was fully evaluated at the editorial level and by three independent peer reviewers.

The reviewers of your manuscript were in general agreement that insight into the mechanisms of the Francisella T6SS would contribute important knowledge to the field and address a significant gap in our understanding of Francisella pathogenesis. However, the reviewers raised some substantial concerns about the manuscript in its current form and agreed that improvements should be made prior to PLOS Pathogens considering a revised version of your study.  These issues must be addressed before we would be willing to consider a revised version of your study. We therefore ask you to modify the manuscript according to the review recommendations before resubmitting your manuscript.

Your revision should address the specific points made by each reviewer. Note that we may send your revised manuscript out for more critical reviews, should we find that additional expertise is necessary to judge the integrity of any additional experiments. Please pay particular attention to the following reviewer suggestions and give them due consideration.

The in vitro secretion assays should have controls where a critical component of the T6SS (not ClpB) is deleted so that T6SS-dependent IglC secretion can be assessed.Address the discrepancies between experiments where growth in vitro and in vivo can or cannot be rescued by plasmid-encoded DNTD ClpBThe manuscript would dramatically benefit by highlighting the differences between ClpV and ClpB with respect to their binding VipB and IglB, respectivelyAttempts to complement ClpB deletion in F. tularensis with Hsp104 should be properly controlled or omitted.There are points in the manuscript that refer to the ATPase activity of ClpB, but the data are not shown.

(1) A letter containing a detailed list of your responses to the review comments and a description of the changes you have made in the manuscript.

(2) Two versions of the manuscript: one with either highlights or tracked changes denoting where the text has been changed; the other a clean version (uploaded as the manuscript file).

Additionally, to enhance the reproducibility of your results, PLOS recommends that you deposit your laboratory protocols in protocols.io, where a protocol can be assigned its own identifier (DOI) such that it can be cited independently in the future. For instructions see http://journals.plos.org/plospathogens/s/submission-guidelines#loc-materials-and-methods

We hope to receive your revised manuscript within 60 days. If you anticipate any delay in its return, we ask that you let us know the expected resubmission date by replying to this email. Revised manuscripts received beyond 60 days may require evaluation and peer review similar to that applied to newly submitted manuscripts.

[LINK]

Sincerely,

Aria Eshraghi

Guest Editor

PLOS Pathogens

Nina Salama

Section Editor

PLOS Pathogens

Kasturi Haldar

Editor-in-Chief

PLOS Pathogens

orcid.org/0000-0001-5065-158X

Grant McFadden

Editor-in-Chief

PLOS Pathogens

orcid.org/0000-0002-2556-3526

Reviewer's Responses to Questions

**Part I - Summary**

Reviewer #1: In the paper by Alam et al., the authors disentangle the role of ClpB in virulence of Francisella. ClpB is known to be involved in disaggregation of inactivated proteins as well as in disassembly of contracted T6SS sheaths. The authors identified key residues responsible for interaction between ClpB and DnaK and show that this interaction was not important for T6SS activity and virulence but was needed for recovery of the bacteria from heat shock. In addition, the N-terminal domain of ClpB was required for T6SS function but not for heat shock. This publication nicely shows that ClpB has two distinct roles and identifies that only its effect on T6SS dynamics is required for virulence. The manuscript is well written and data are well presented except few cases mentioned bellow.

Reviewer #2: Type VI secretion systems (T6SSs) play crucial roles in bacterial pathogenicity and bacterial warfare by delivering toxins into target cells. This system is energized by the contraction of a huge macromolecule (e.g. IglB, VipB), which needs to be disassembled afterwards to allow for a new secretion cycle. This activity is typically executed by ClpV, a member of the AAA+ protein family. Francisella tularensis encodes for a modified T6SS that is lacking ClpV. It was shown by the authors and others before that ClpV function is taking over by ClpB, another AAA+ protein. ClpB typically acts in protein disaggregation and provides heat resistance to bacteria. This function of ClpB requires cooperation with the bacterial Hsp70 chaperone (DnaK). ClpB directly binds to DnaK via a coiled-coil domain (M-domain) and M-domain mutations abrogating DnaK interaction are non-functional in disaggregation.

Here, the authors analyzed whether the function of F. tularensis ClpB requires DnaK activity. They convincingly show that ClpB acts independently as M-domain mutants that are deficient in mediating heat tolerance and disaggregation are still functional in T6SS. Additionally, the authors document a critical role of the N-terminal domain (NTD) of ClpB, which likely mediates binding to the T6SS substrate (IglB).

The study is well performed and provides novel insights into the mechanism of T6SS, warranting publication in PLOS Pathogens. However, there are a couple of issues, which need to be addressed in a revised version as detailed below. In particular, the ClpB ATPase mutant proteins require further biochemical validation and the binding specificity of the ClpB NTD should be explored/discussed in more detail.

Reviewer #3: The manuscript by Alam et al aims to answer questions about the role of the Francisella tularensis ClpB chaperone in helping to assemble/disassemble the T6SS, including by interacting with the chaperone DnaK/Hsp70. The authors first modeled ClpB-DnaK interactions to identify potential amino acid residues of interest (notably ClpB Y503), then directly examined the roles of individual residues in ClpB-DnaK interactions by site directed mutagenesis and protein disaggregration, protein refolding, and heat shock tolerance studies. However, despite ClpB Y503 preventing association with DnaK, ClpB Y503 mutants retained virulence in macrophages and mice and secreted IglC (a surrogate for T6SS formation), indicating that T6SS/virulence are not affected by the clpB Y503 mutation (DnaK interaction). Interestingly, these ClpB Y503 mutants were heat sensitive, indicating that ClpB-DnaK interactions only are important for heat shock. The authors next generated single and double mutants of the putative nucleotide binding domains (NDBs) in ClpB, finding that single or double mutations of the NBDs reduced macrophage replication, heat shock tolerance, T6 secretion, and virulence in mice. Finally, the authors deleted the N-terminal 156 amino acids of ClpB, finding that the N-terminal region was important, but not essential, for heat shock survival, replication in macrophages, and virulence in mice. Either the N-termal ClpB region from E. coli or the entire ClpB protein from E. coli could complement ClpB mutants. There is a substantial amount of data provided in this manuscript and this study provides new information about how the F. tularensis ClpB protein contributes to heat shock tolerance, is involved in T6 secretion of virulence factors, is required for virulence in macrophages and mice, and that specific amino acid residues are required for DnaK interactions and (presumably) ATP binding and hydrolysis. However, it is important to note that ATP binding and hydrolysis is purely speculative as the authors did not directly test this.

**Part II – Major Issues: Key Experiments Required for Acceptance**

Reviewer #1: (No Response)

Reviewer #2: 1. The biochemical characterization of the ClpB mutants is in parts incomplete. The authors need to document the oligomerization behavior of the variants, as hexamer formation is critical for ClpB function. This has not been done for ATPase mutants and is only mentioned as “data not shown” (page 8, line 166) for M-domain mutants. Similarly, the authors did not determine ATPase activities of all ClpB mutants. Most mutant proteins have been purified and therefore completion of the biochemical characterization seems doable.

2. Figure 5D: The defect of F. tularensis ΔclpB cells in intracellular replication can be rescued by plasmid-encoded ΔNTD-ClpB. This result is surprising and is in contradiction to the crucial function of the ClpB NTD in T6SS and F. tularensis proliferation in mice (Fig. 5C/E) but also to Fig. 3A/4A, which document a crucial function of ClpB in the very same assay. The authors need to comment on this unexpected result.

3. The ClpV NTD harbors an additional α-helix that is crucial for binding to the T6SS substrate. This helix is missing in ClpB, suggesting that the interaction details of ClpV and ClpB with VipB/IglB substrates are different. The authors shortly report on this aspect in the discussion section (page 16, lines 365-367), but a more detailed analysis and comparison of the substrates (IglB: ClpB; VipB: ClpV) is required here. It is recommended to at least show sequence alignments of IglB and VipB homologs and to highlight the differences in their N-terminal regions (which in case of VipB includes the binding site for ClpV).

4. Figure 6: The authors show that the yeast ClpB homolog, Hsp104, cannot function in F. tularensis T6SS. The molecular reason remains unclear as Hsp104 shares substantial sequence homology with ClpB including the NTD. Furthermore, the authors did not determine Hsp104 expression levels, leaving the possibility that Hsp104 is inactive due to poor expression. The presented data therefore do not add much to the manuscript and should be removed.

Reviewer #3: line 110. The authors begin the results section by describing how they looked for potential interactions between ClpB and the DnaK/Hsp70 chaperone system but no background information was provided for the DnaK/Hsp70 chaperone system, which actually is complicated - including DnaK (Hsp70), DnaJ, and GrpE. Please explain. More importantly, it is not clear why DnaK/Hsp70 is being examined in this study, compared to DnaJ or GrpE.

line 149-153. The authors note that Q502A showed very most reactivation (19%), while Y503A showed only background activity. However, Figure 2A appears to indicate little difference between KJE alone, Y503A, and Q502A. How can Y503A be background but Y503A has a measurable difference?

line 162-163. The authors note that the “disaggregation activities of E500A (61%) and E508A (68%) were … not much lower than that of wild-type ClpB (83%).” However, this reviewer believes that 15% and 22% differences are substantial.

line 219-220. It is not clear from S1A Fig, Figure 4, or the materials and methods what specifically was done to generate single or double mutants within both NBDs. Supplemental Table S1 provides this information but it is not immediately clear what was done. Please indicate here so readers don’t have to dig through supplemental material.

lines 236-246 and Figure 4C. It is not clear what point the authors are trying to make about higher secretion levels in WA2, WB1, WB2, and Arg2…is this relevant, substantial, and/or significant? The last sentence about structural topology and protein stability of double mutants also is not clear, given ambiguous data in Figure 4C. All of this needs to be clarified.

line 247-251. Perhaps the largest flaw of the paper is in this section, where the authors note ‘the inhibition of ATP binding and hydrolysis….” Where is the data specifically showing ATP binding to ClpB and these ClpB mutants? Where is the data specifically measuring ATP hydrolysis? Totally aside from ATP binding and hydrolysis, why didn’t the authors generate other ClpB mutants (mutating 4 amino acids or 8 amino acids) outside of the NBD domain and test these mutants in these same assays? How are the authors sure that these specific regions (WA1, WB1, Arg1, etc.) actually account for defects in shock survival, intracellular replication, and T6 secretion?

line 274-283. Given the substantial amount of modeling and domain predictions in other parts of the paper (S1A and Fig 5), it is surprising that the authors decided to delete the N-terminal 156 amino acids of ClpB (i.e. a major percentage of the ClpB protein)

**Part III – Minor Issues: Editorial and Data Presentation Modifications**

Reviewer #1: Line 74 – citation “14” is wrong for this statement. The correct would be: Basler et al., Nature 2012 and Basler & Mekalanos, Science 2012.

Line 369 – citation “37” is wrong for this statement (see above).

Line 353 - wrong citation.

Figure 2A,B – show error bars and averages rather than representative data.

Figure 3B - Do authors have a control where a critical component of T6SS was deleted? To know if the observed amount of IglC in the supernatant of ClpB-negative cells appeared there in a T6SS dependent manner.

Figure S4 - labels missing.

Reviewer #2: 1. page 7, line 149: In Fig. 2A the authors determine the disaggregation activity of ClpB (wild type and mutants) by monitoring the decrease of Malate Dehydrogenase aggregate turbidity. The term “reactivation” is therefore incorrect for this experimental setting.

2. Page 7, line 154: The term “re-aggregation” activity of Y503A is unclear and needs to be defined.

3. Suppl. Fig. 3B: It is unclear which data points are depicted here. Do they belong to ClpB wild type and/or mutant proteins?

Reviewer #3: line 109-110: should be “however, the interaction between ClpB and…”

Fig 5A vs. S1A. Fig 5A shows R503 but no R is present in at position 503 in S1A. Similarly, Fig 5A shows T556 but no T is present at position 556 in S1A. Why doesn’t Fig 5A include relative positions of Arg1 and Arg2?

PLOS authors have the option to publish the peer review history of their article (what does this mean?). If published, this will include your full peer review and any attached files.

Reviewer #1: No

Reviewer #2: No

Reviewer #3: Yes: Jason F. Huntley

---

## [Editor Report · Decision Letter 1]

6 Mar 2020

Dear Dr. Sjostedt,

We are pleased to inform you that your manuscript 'Dissociation between the critical role of ClpB of Francisella tularensis for the heat shock response and the DnaK interaction and its important role for efficient type VI secretion and bacterial virulence' has been provisionally accepted for publication in PLOS Pathogens.

Best regards,

Aria Eshraghi

Guest Editor

PLOS Pathogens

Nina Salama

Section Editor

PLOS Pathogens

Kasturi Haldar

Editor-in-Chief

PLOS Pathogens

orcid.org/0000-0001-5065-158X

Michael Malim

Editor-in-Chief

PLOS Pathogens

orcid.org/0000-0002-7699-2064

Dear Dr. Sjostedt,

Thank you for resubmitting your revised manuscript "Dissociation between the critical role of ClpB of Francisella tularensis for the heat shock response and the DnaK interaction and its important role for efficient type VI secretion and bacterial virulence" (PPATHOGENS-D-19-01936R1). I have reviewed your response to the reviewers and find that you have made adequate changes to address their concerns. Congratulations on a thorough and insightful manuscript. I look forward to seeing it published.

Best regards,

Aria Eshraghi
---

## [Editor Report · Acceptance letter]

2 Apr 2020

Dear Dr. Sjostedt,

We are delighted to inform you that your manuscript, "Dissociation between the critical role of ClpB of *Francisella tularensis* for the heat shock response and the DnaK interaction and its important role for efficient type VI secretion and bacterial virulence," has been formally accepted for publication in PLOS Pathogens.

Best regards,

Kasturi Haldar

Editor-in-Chief

PLOS Pathogens

orcid.org/0000-0001-5065-158X

Michael Malim

Editor-in-Chief

PLOS Pathogens

orcid.org/0000-0002-7699-2064